# Radixin modulates the function of outer hair cell stereocilia

Sonal Prasad [1 ✉], Barbara Vona [2], Marta Diñeiro[3], María Costales[4], Rocío González-Aguado [5], Ana Fontalba[6], Clara Diego-Pérez[7], Asli Subasioglu[8], Guney Bademci[9], Mustafa Tekin[9,10,11], Rubén Cabanillas[12], Juan Cadiñanos [3] & Anders Fridberger [1 ✉]

The stereocilia of the inner ear sensory cells contain the actin-binding protein radixin, encoded by *RDX*. Radixin is important for hearing but remains functionally obscure. To determine how radixin influences hearing sensitivity, we used a custom rapid imaging technique to visualize stereocilia motion while measuring electrical potential amplitudes during acoustic stimulation. Radixin inhibition decreased sound-evoked electrical potentials. Other functional measures, including electrically induced sensory cell motility and sound-evoked stereocilia deflections, showed a minor amplitude increase. These unique functional alterations demonstrate radixin as necessary for conversion of sound into electrical signals at acoustic rates. We identified patients with *RDX* variants with normal hearing at birth who showed rapidly deteriorating hearing during the first months of life. This may be overlooked by newborn hearing screening and explained by multiple disturbances in postnatal sensory cells. We conclude radixin is necessary for ensuring normal conversion of sound to electrical signals in the inner ear.

[1] Department of Biomedical and Clinical Sciences, Linköping University, SE-581 83 Linköping, Sweden. [2] Department of Otorhinolaryngology, Head and Neck Surgery, Tübingen Hearing Research Centre, Eberhard Karls University Tübingen, 72076 Tübingen, Germany. [3] Laboratorio de Medicina Molecular, Instituto de Medicina Oncologica y Molecular de Asturias, 33193 Oviedo, Spain. [4] Department of Otorhinolaryngology, Hospital Universitario Central de Asturias, 33011 Oviedo, Spain. [5] Department of Otorhinolaryngology, Hospital Universitario Marqués de Valdecilla, 39008 Santander, Spain. [6] Department of Genetics, Hospital Universitario Marqués de Valdecilla, 39008 Santander, Spain. [7] Department of Otorhinolaryngology, Hospital Universitario de Salamanca, 33007 Salamanca, Spain. [8] Department of Medical Genetics, Izmir Ataturk Education and Research Hospital, Izmir 35360, Turkey. [9] John P. Hussman Institute for Human Genomics, University of Miami Miller School of Medicine, Miami, FL 33136, USA. [10] Department of Otolaryngology, University of Miami Miller School of Medicine, Miami, FL 33136, USA. [11] Dr. John T. Macdonald Department of Human Genetics, University of Miami Miller School of Medicine, Miami, FL 33136, USA. [12] Área de Medicina de Precisión, Instituto de Medicina Oncologica y Molecular de Asturias, 33193 Oviedo, Spain. ✉email: sonal.prasad@liu.se; anders.fridberger@liu.se

The sensory cells of the inner ear are equipped with stereocilia, which harbor the molecular machinery that permits sound to be converted into electrical potentials. The protein radixin appears to be an important component of this machinery, since radixin-deficient mice are deaf[1] from an early age and biallelic variants in the human *RDX* gene is a cause of non-syndromic neurosensory hearing loss (DFNB24; MIM #611022[2,3]. From these observations, it is clear that radixin is necessary for normal hearing, but the physiological role of the protein remains obscure.

Radixin is enriched within stereocilia[4], and bioinformatic analyses suggest that it is a hub in a network of interacting molecules[5] associated with the mechanotransduction process, such as phosphatidylinositol-4,5-bisphosphate (PIP$_2$), calmodulin, and calcium[6]. While the functional relevance of these interactions has not been clarified, it is evident that phosphorylated radixin links the actin cytoskeleton with various transmembrane adhesion proteins, such as CD44[7,8].

Given radixin's important role in the network of proteins within stereocilia, we hypothesized that radixin may contribute to the regulation of stereocilia function in the mature inner ear. Little is known about this regulation, but we note that stereocilia may be capable of active force generation[9,10], acting in concert with forces generated within the soma of outer hair cells[11,12] to establish normal hearing sensitivity and frequency selectivity.

To determine the influence of radixin on cochlear amplification and sensory cell function, we used a custom rapid confocal imaging technique to examine stereocilia motion while recording the electrical potentials produced by the sensory cells during acoustic stimulation. These measurements revealed an unusual pattern of functional changes when radixin was disabled. The sound-evoked electrical potentials were substantially reduced despite other important functional measures, such as stereocilia deflections and electrically induced motility being intact. This shows that radixin is necessary for the normal function of mechanically sensitive ion channels, allowing them to work at acoustic rates.

We also provide a clinical characterization of patients with *RDX* variants. Their hearing was normal early in life, presumably because ezrin partially substitutes for radixin, but hearing was lost during the first months of life. This causes a delay in diagnosis, and also means that a brief therapeutic window exists in the event that specific therapies aimed at DFNB24 become available.

## Results

### Clinical findings in patients with biallelic variants in the *RDX* gene. The first patient was a 2-year old girl of Moroccan origin born to term after a normal pregnancy and delivery. Maternal serology was positive for rubella and negative for hepatitis B, human immunodeficiency virus, *Toxoplasma gondii*, and syphilis, ruling out these agents as contributors to hearing loss. There was no risk for chromosomal abnormalities or metabolopathies, as revealed by standard screening. The only risk factor was consanguinity, as her parents were cousins. Importantly, hearing screening before the third day of life revealed that otoacoustic emissions, faint sounds produced by the inner ear in response to low-level acoustic clicks, were present. Since the sensory outer hair cells must be intact for otoacoustic emissions to be generated, this ruled out peripheral hearing loss[13].

However, at the age of 16 months, the patient was referred to the ENT department because of suspected hearing loss. Otoacoustic emissions could not be detected, suggesting that peripheral hearing loss had developed. Auditory evoked potentials were absent and steady-state evoked potential testing revealed a bilateral threshold of 90 dB hearing level at 0.5 and 1 kHz (a pedigree and the patient's evoked potential audiogram are shown in Fig. 1a). These findings are diagnostic of profound hearing loss.

Genetic testing with the OTOgenics panel[14] revealed a homozygous alteration in the *RDX* gene (NM_002906.3: c.129 G > A, p.W43X), which was confirmed with Sanger sequencing. The result is a truncation of the protein in exon 3 (of 14), leaving only a part of the membrane-binding domain but stripping all of the actin-binding C-terminus, a change that completely disables radixin since most of its length is lost.

The second patient was female and adopted at 6 months of age. Early hearing screening was performed with brainstem auditory-evoked potentials and the patient passed. However, she was referred to the ENT department at 8 months of age with a suspicion of hearing loss. Testing with steady-state evoked potentials showed moderate hearing loss (Fig. 1b). Genetic testing indicated a homozygous deletion of all of *RDX*'s second exon, where the initiation codon is located (NM_002906.3: c.-64-1215_12 + 348del). Notably, an in-frame start codon present in exon 3 may mean that a protein 11 amino acids shorter is present in this patient. This shortened protein should be capable of attaching to the actin cytoskeleton, but membrane binding will be affected.

Our third case was diagnosed with hearing loss in infancy and underwent pure-tone audiometry at the age of 8 years, revealing a bilaterally symmetrical profound sensorineural hearing loss (Fig. 1c). Exome sequencing disclosed a homozygous nonsense variant in exon 11 of *RDX* (NM_002906.3: c.1108 C > T, p. R370X). This removes the highly conserved actin-binding motif (exons 13 and 14)[2], preventing radixin from interacting with actin filaments. Otoacoustic emissions were absent, and a younger sister was similarly affected with symmetrical profound sensorineural hearing loss without otoacoustic emissions. Both siblings had the same homozygous *RDX* variant. Neonatal hearing screening was not performed in either case.

These clinical data show that patients with *RDX* alterations that result in a non-functional protein can pass newborn hearing screening on the first days of life, but hearing sensitivity deteriorates thereafter. It is not clear why this hearing loss develops, so we performed experiments to determine the functional role of radixin.

### Radixin expression in the hearing organ. To study radixin's influence on hearing and the role of the protein for stereocilia function (Fig. 2a), we used temporal bone preparations isolated from guinea pigs, a species with low-frequency hearing similar to humans. In these isolated preparations (Fig. 2b), which retain the passive mechanics of the hearing organ[15], direct visualization of sound-evoked stereocilia motion is possible in a nearly native environment (Fig. 2c, d)[16], which makes the preparation useful for investigating functional changes in stereocilia. However, the presence and distribution of radixin has not previously been examined in guinea pig hair cells, so we began by staining the mature organ of Corti with phosphospecific antibodies targeting radixin's threonine 564 residue. Double-labeling with fluorescently tagged phalloidin, which binds actin filaments (Fig. 2e), was used to locate stereocilia.

The strongest fluorescence was observed in the three rows of outer hair cells, which were intensely labeled by radixin antibodies (Fig. 2e). Labeling of inner hair cell stereocilia was less prominent, and no consistent radixin label was present in either the cell bodies of the sensory cells, in their adjacent supporting cells, or in the synaptic regions of the inner hair cells (Fig. 2f).

Three-dimensional reconstructions of outer hair cell stereocilia (Fig. 2g) showed that radixin labeling was most intense in the

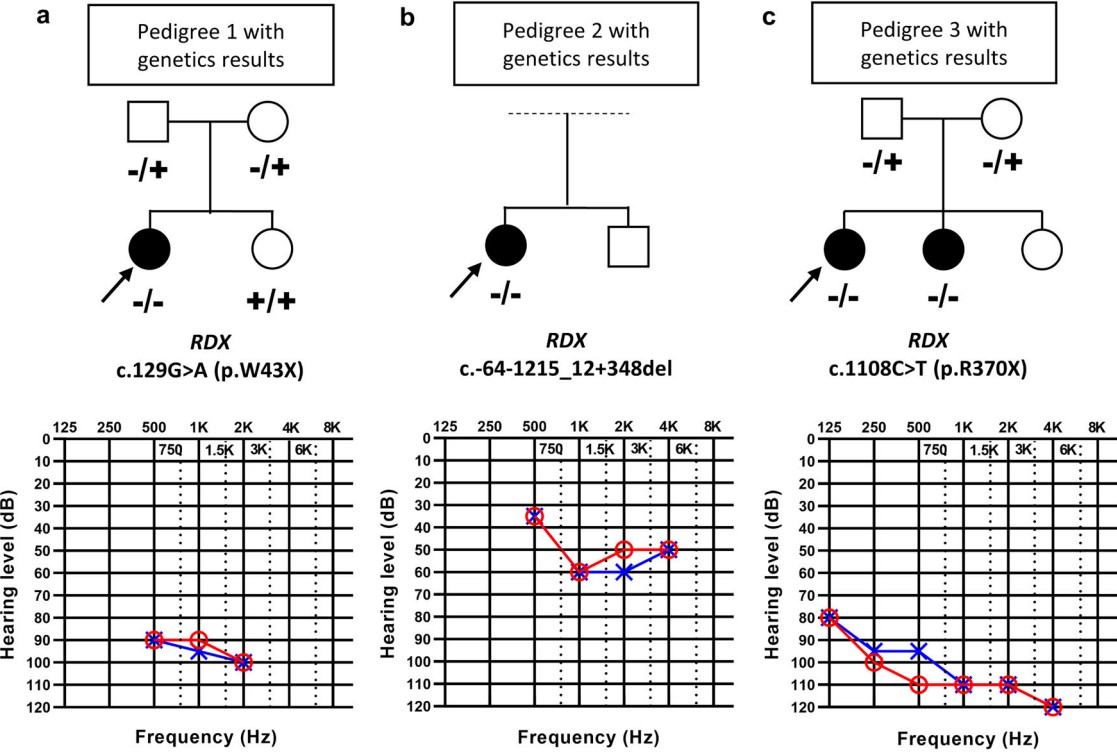

**Fig. 1 Hearing impairment in patients with *RDX* variants.** Pedigrees of the families with non-syndromic sensorineural hearing loss (a: patient I, b: patient II, c: patient III and IV). The probands are shown with arrows. Open symbols: unaffected; filled symbols: affected. Audiograms and steady-state evoked potentials showed different degrees of bilateral sensorineural hearing loss of affected individuals (red, right ear; blue, left ear). **a** Steady-state evoked potentials revealed profound bilateral hearing loss in patient I at 16 months of age. **b** Steady-state evoked potentials in patient II showed moderate bilateral sensorineural hearing loss at 8 months of age. This patient was reported in a Spanish hearing impaired cohort genetic study[14]. **c** Audiogram of patient III showed profound bilateral sensorineural hearing loss at 8 years of age. The variant found in this patient was included in a list of variants in hearing-loss genes[40], but no information about the patient was provided. Hearing thresholds of all four patients show a sloping configuration ranging from mild (patient II) to severe (patients I, III and IV) sensorineural hearing loss at low frequencies and profound impairment at high frequencies.

mid-basal part of stereocilia and tapered off toward their tip. To quantify this more precisely, we measured the fluorescence intensity of each probe as a function of distance from the base of the hair bundle. Plots of the normalized fluorescence profiles (Fig. 2h) confirmed the stronger labeling near the base of stereocilia, unlike the actin probe (phalloidin), which had similar labeling intensity through the length of the hair bundle.

Since the actin probe had stronger emission, we were concerned that its fluorescence might bleed through into the radixin channel. If this were the case, a linear relationship between their fluorescence intensities is expected. However, no such relationship was found (Fig. 2i).

The radixin labeling pattern is consistent with the one found in chick[17] and rat[18] inner ears, so we conclude that guinea pigs are an adequate model for investigating the functional role of radixin in the mature hearing organ. Next, we performed physiological measurements by combining rapid confocal imaging of sound-evoked stereocilia motion with electrophysiology, measurements of electrically evoked motion, fluorescence recovery after photobleaching (FRAP), and in vivo measurements of hearing sensitivity in animals whose inner ears were treated with radixin inhibitors.

**Radixin inhibition influences stereocilia deflections.** Having established that radixin is present in guinea pig hair cells, but not detectable in supporting cells or in afferent neurons, we proceeded by examining the sound-evoked responses of stereocilia. To label stereocilia, a double-barreled glass microelectrode with 3-μm tip diameter was positioned close to the sensory cells. One

electrode barrel was used for introducing the fluorescent dye di-3-ANEPPDHQ, which stained stereocilia (see also Fig. 2c) and allowed their sound-evoked motion to be studied using time-resolved confocal imaging[19,20]. The other electrode barrel was used for delivering the radixin blocker DX-52-1, which disrupts radixin's ability to link the actin cytoskeleton with the cell membrane[21,22] (Fig. 2a; DX-52-1 also has weak inhibitory effect on ezrin and moesin[22], which are not detectable in the mature organ of Corti[1], and a weak blocking effect on galectin-3[23]. Galectin-3 knockout mice have normal hearing and normal acoustic startle responses[24], so inhibition of this protein is not expected to affect organ of Corti function. DX-52-1 was also evaluated in neuronal cell lines, where an effect on cell motility was found only in cells expressing radixin[25]. The loss of the membrane–cytoskeleton interactions creates an effect similar to the truncating mutations described in our patients.

After injecting a 1-mM solution of DX-52-1 dissolved in artificial endolymph, no morphological changes were observed in stereocilia (except for minor alterations in the brightness of the dye, Fig. 3a; note that the effective inhibitor concentration is reduced because the injected solution is dissolved in the endolymph present in scala media), but the injection changed the response to acoustic stimulation. Before DX-52-1 (Fig. 3b), the base of the hair bundle (blue trajectory) had a different direction of motion than the bundle tip (red trajectory). As a result of this difference, motion directed at scala tympani (downwards in Fig. 3b) caused deflection of stereocilia toward the center of the cochlear spiral (green trajectory). Ten to fifteen minutes after DX-52-1 (Fig. 3b), sound-evoked displacement

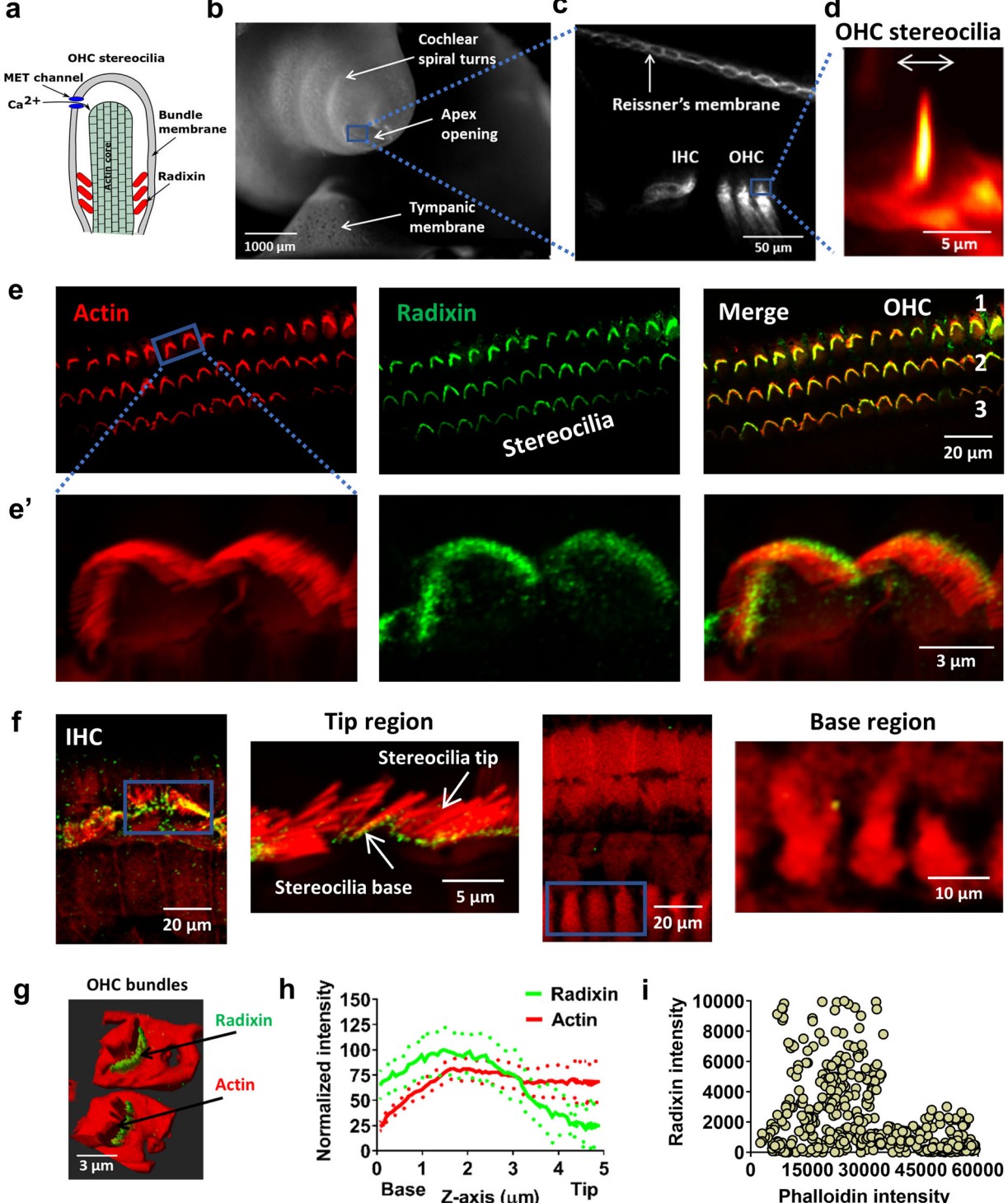

**Fig. 2 Radixin expression and localization in guinea pig cochlear hair cells. a** Schematic diagram showing the putative function of radixin in stereocilia. **b** A low magnification image of the temporal bone preparation. Note the apical opening used for imaging. **c** Release of the dye di-3-ANEPPDHQ into the endolymphatic space stained Reissner's membrane as well as the hair bundles. **d** Outer hair cell (OHC) stereocilia imaged during sound stimulation at 220 Hz, 80 dB sound pressure level. **e** Representative confocal images of sections of the organ of Corti labeled with phalloidin (red, staining actin) and a radixin-specific monoclonal antibody (green), as well as an overlay. The bundles of the sensory hair cells are intensely labeled by the radixin antibody. OHC 1, 2, 3 indicate the three rows of outer hair cells. Images were taken from the surface preparations of the apical turn. **e'** Inset showing a higher magnification view. **f** Three-dimensional reconstruction of the inner hair cell area shows absence of radixin label in the cell bodies of the inner hair cells. Likewise, no radixin label was detected in the neuronal or synaptic region of the inner hair cells. **g** A 3D reconstruction of OHC stereocilia showing predominance of radixin labeling near the stereocilia base and consistent actin labeling in the cuticular plate and stereocilia bundles. **h** Normalized average ± s.e.m. (dotted) signal intensity profiles for radixin and actin expression (average of 11 bundles from 3 different animals) which shows decline in radixin labeling toward the tip of stereocilia and consistent actin labeling by phalloidin throughout the stereocilia. **i** Scatter plot showing lack of relation between radixin and phalloidin pixel intensities.

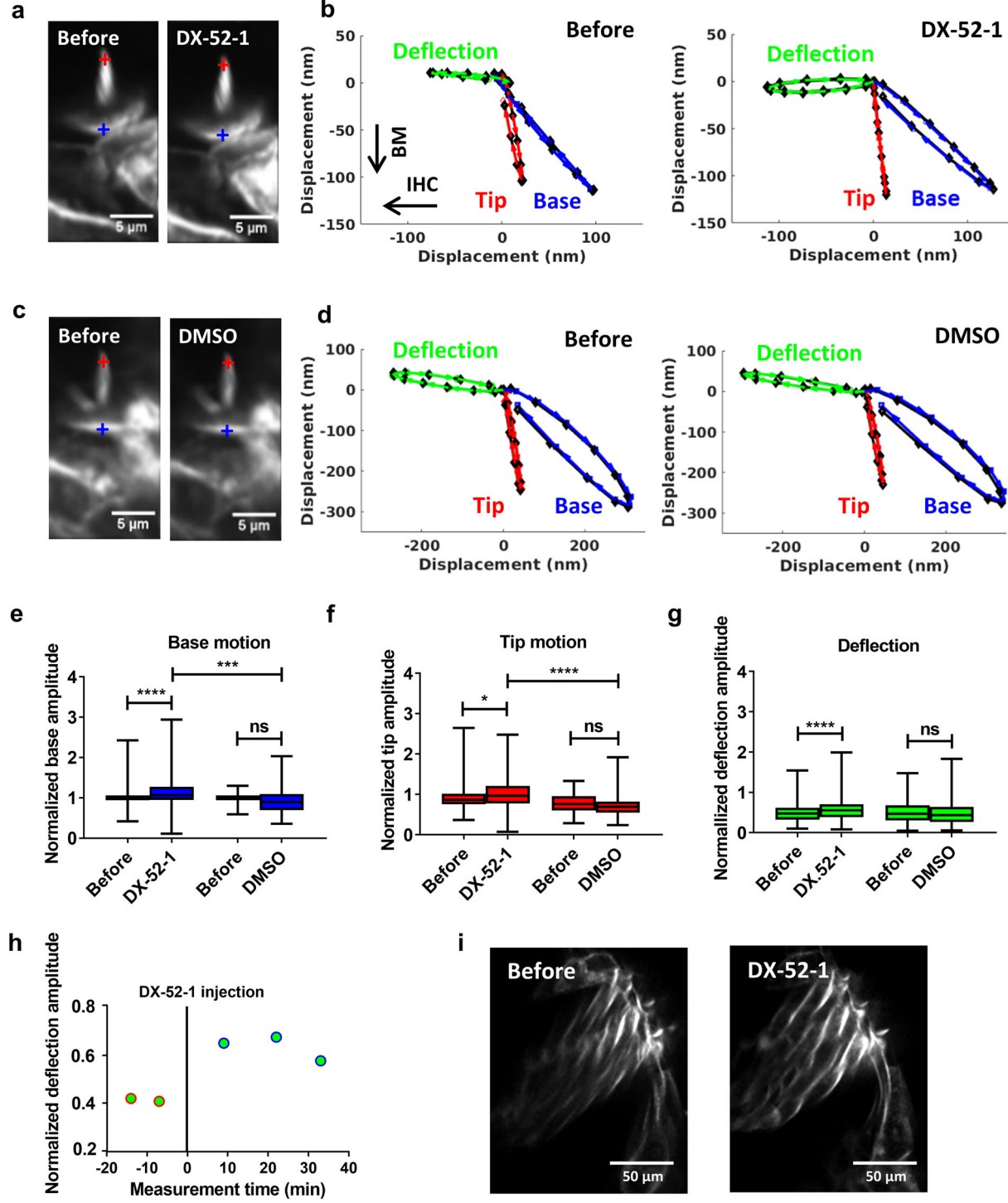

**Fig. 3 DX-52-1-induced effects on sound-evoked motions of outer hair cell stereocilia. a** Time-resolved confocal images acquired during sound stimulation show that DX-52-1 does not alter the morphology of stereocilia (except for a small change in the brightness of the fluorescent dye). **b** Sound-evoked motion of the bundle tip (red) and base (blue) before and after DX-52-1 injection in an example preparation. The stimulus was a pure tone at 220 Hz and 80 dB sound pressure level. By subtracting trajectories from the tips and bases of stereocilia, a measure of the deflection of the bundle is obtained (green). **c** Confocal image obtained after DMSO injection, showing lack of effect on stereocilia morphology. **d** No change in the motion of the bundle tip (red) and base (blue) before and after DSMO injection observed along with absence of change in deflection (green). **e–g** Averaged bundle motion change at the base of outer hair cell stereocilia (blue bar), at their tip (red bar) and the deflection of the bundle (green bar). Data were normalized to the base trajectory amplitude recorded before the injection. Averaged data from 70 (DX-52-1) and 15 (DMSO) individual preparations ±standard deviation. **h** Time course of deflection amplitude of outer hair cell bundles (blue circle) in an example preparation. The vertical line at time zero indicates the time of injection. Data were normalized to the average trajectory amplitude recorded before injection. **i** High-resolution confocal images of stereocilia in preparations treated with DX-52-1 or vehicle show no detectable morphological changes. ****$P < 0.0001$; ***$P < 0.001$; **$P < 0.01$; *$P < 0.05$; n.s., not significant; two-tailed paired $t$-test, two-tailed unpaired $t$-test with Welch's correction.

showed a minor but significant increase both at the base and at the tip of the hair bundle. As a consequence, the sound-evoked deflection of the hair bundle became larger (green trajectory in Fig. 3b). In preparations treated with vehicle alone (endolymph with 1.8% Dimethyl sulfoxide, DMSO), neither morphology nor motion trajectories changed (Fig. 3c, d).

Figure 3e–g shows the hair bundle motion change across 70 preparations. At both the tip and the base of the hair bundle, the motion amplitude increased (from $98 \pm 15$ nm to $116 \pm 48$ nm at the base; $P = 0.00001$, two-tailed paired $t$-test, Fig. 3e; and from $90 \pm 24$ nm to $102 \pm 40$ nm at the tip; $P = 0.04$, Fig. 3f). Base motions were larger than the tip motion[26]. The change in the deflection amplitude was also significant (from $48 \pm 21$ nm to $56 \pm 28$ nm; $P = 0.00001$, two-tailed paired $t$-test; Fig. 3g). A significant difference was also found when preparations injected with DX-52-1 were compared to those injected with vehicle alone (Fig. 3e, f; $n = 27$; two-tailed unpaired $t$-test with Welch's correction). The change induced by DX-52-1 was apparent 10 min after its application, deflections remained elevated for at least 10 min, and a gradual recovery took place thereafter (Fig. 3h). Although the stereocilia showed no overt signs of damage after DX-52-1 (e.g., Fig. 3a), we nevertheless performed a separate set of experiments to image stereocilia at higher resolution after DX-52-1. These experiments showed no morphological alterations to stereocilia 30 min after DX-52-1 (Fig. 3i).

We also evaluated the effect of DX-52-1 on inner hair cell stereocilia. However, inner hair cells are less numerous than outer hair cells and their stereocilia often do not label as well with di-3-ANEPPDHQ and are slightly damage-prone. In a more limited sample of inner hair cells, there was no consistent effect of DX-52-1 on sound-evoked stereocilia motion (Supplementary Fig. 1a–e, $n = 18$).

In summary, the data shown in Fig. 3 demonstrate that the radixin blocker DX-52-1 affected the sound-evoked motion of stereocilia, causing mildly increased deflection amplitudes. This finding clearly cannot explain the hearing loss seen in patients, but it is consistent with an effect of radixin on the stiffness of stereocilia.

**Radixin affects electrically evoked motility.** Outer hair cells contain a transmembrane protein, prestin, which causes rapid changes of cell length in response to alterations in membrane potential[11]. This electromotility is critical for hearing, and to further probe radixin's influence on hair cell function, we measured electrically evoked motility using the rapid imaging technique described above. The double-barreled microelectrode allowed us to apply 10-µA square wave currents at the frequency of 5 Hz. These currents changed the electrical potential in scala media, resulting in increased currents through the MET channels and increased force production by outer hair cells[27].

To show the change in electromotility evoked by DX-52-1, Fig. 4a shows an outer hair cell imaged in situ during electrical stimulation. The green channel was acquired during the negative part of the square wave and the red channel during its positive phase. Before DX-52-1 application, most pixels overlapped, signifying low motility amplitude (Fig. 4a). After DX-52-1 was introduced, the green and the red channels separated, implying an increased amplitude of electromotility (Fig. 4a). These changes were quantified through optical flow analysis. The time course (Fig. 4b) shows that the increase was evident 10 min after injection of the blocker. A tendency to recovery was seen thereafter. Overall, the change induced by DX-52-1 was statistically significant (from $101 \pm 22$ nm to $139 \pm 135$ nm; $P = 0.001$, two-tailed paired $t$-test; $n = 70$), but this was not the case in preparations injected with the vehicle alone (Fig. 4c), where no significant change in motion occurred.

Electrically evoked motility requires currents to pass through stereocilia and into the cell bodies of the outer hair cells. Since we found an increased amplitude of electrically evoked motion, these channels remained able to pass low-frequency currents during DX-52-1 application. The increase in electromotility is consistent with a slightly decreased organ of Corti stiffness, in agreement with the changes in sound-evoked stereocilia motion described above. However, neither finding explains why hearing is impaired in patients with *RDX* variants.

**The site of action of DX-52-1 is the stereocilia.** To verify that DX-52-1 acts at the level of the stereocilia, we exploited the fact that radixin connects the cell membrane with the underlying actin cytoskeleton. Hence, inhibition of radixin is expected to remove an obstacle to diffusion, increasing the mobility of membrane lipids. Lipid mobility can be measured using FRAP[28]. In brief, a laser beam was focused to a submicron spot to bleach a region of interest on the stereocilia (Fig. 4d). Since diffusion will add new dye molecules to the bleached area, the gradual recovery of fluorescence provides a measure of lipid mobility in the membrane, as seen in the graph in Fig. 4e. Here, a single-phase exponential model (black line) was fitted to the averaged fluorescence recovery curve measured before (red open circles) and 10–15 min after DX-52-1 injection (blue circles). The fit parameters revealed significantly faster fluorescence recovery during the 25–30 min that followed inhibition of radixin (Fig. 4f; $22 \pm 11$ s vs. $14 \pm 7$ s; $P = 0.02$, two-tailed paired $t$-test; $n = 24$). Control injections in 14 preparations showed no significant change in the fluorescence recovery time (Fig. 4f). The normalized diffusion time was slightly longer after vehicle injection (Fig. 4f), but this change was not significant.

We also performed FRAP experiments on inner hair cell stereocilia (Supplementary Fig. 1f–h, $n = 20$), on the cell bodies of the outer hair cells (Fig. 4g–i, $n = 16$) and inner hair cells (Supplementary Fig. 1i–k, $n = 18$), and on the dendrites of the auditory nerve (Supplementary Fig. 2, $n = 18$). In neither case did we find a significant change in the mobility of membrane lipids after DX-52-1 injection, suggesting that this compound specifically affects outer hair cell stereocilia when it is injected in the scala media.

The changes in lipid mobility in the outer hair cells are consistent with disruption of membrane–cytoskeletal interactions when radixin is blocked.

**Radixin inhibition decreased cochlear microphonic potentials.** During sound stimulation, ions permeate mechanically sensitive ion channels from the surrounding fluid, generating extracellular electrical potentials that can be measured through the electrode placed near the sensory cells. By tracking the amplitude of these microphonic potentials over a range of stimulus frequencies, tuning curves were acquired.

Upon injection of DX-52-1, a decrease in the cochlear microphonic (CM) amplitude (Fig. 4j) was evident 10–15 min after the blocker injection, and the amplitude remained depressed during the ensuing 30–35 min (Fig. 4k). On average, the CM amplitude decreased from $124 \pm 87$ µV to $57 \pm 50$ µV, measured at the peak of each tuning curve (Fig. 4l; $P = 0.00001$, two-tailed paired $t$-test; $n = 70$). A significant difference in amplitude was also evident between preparations injected with DX-52-1 and the controls (Fig. 4l; $P = 0.00001$, two-tailed unpaired $t$-test with Welch's correction; $n = 13$ controls).

The decrease in the CM amplitude means that the ability to convert sound into rapidly alternating electrical potentials is impaired. This, however, is not due to a change in the stimulation of stereocilia, because stereocilia deflections were slightly increased (Fig. 3b–i). Also, the decrease in the CM is not due to a blocking

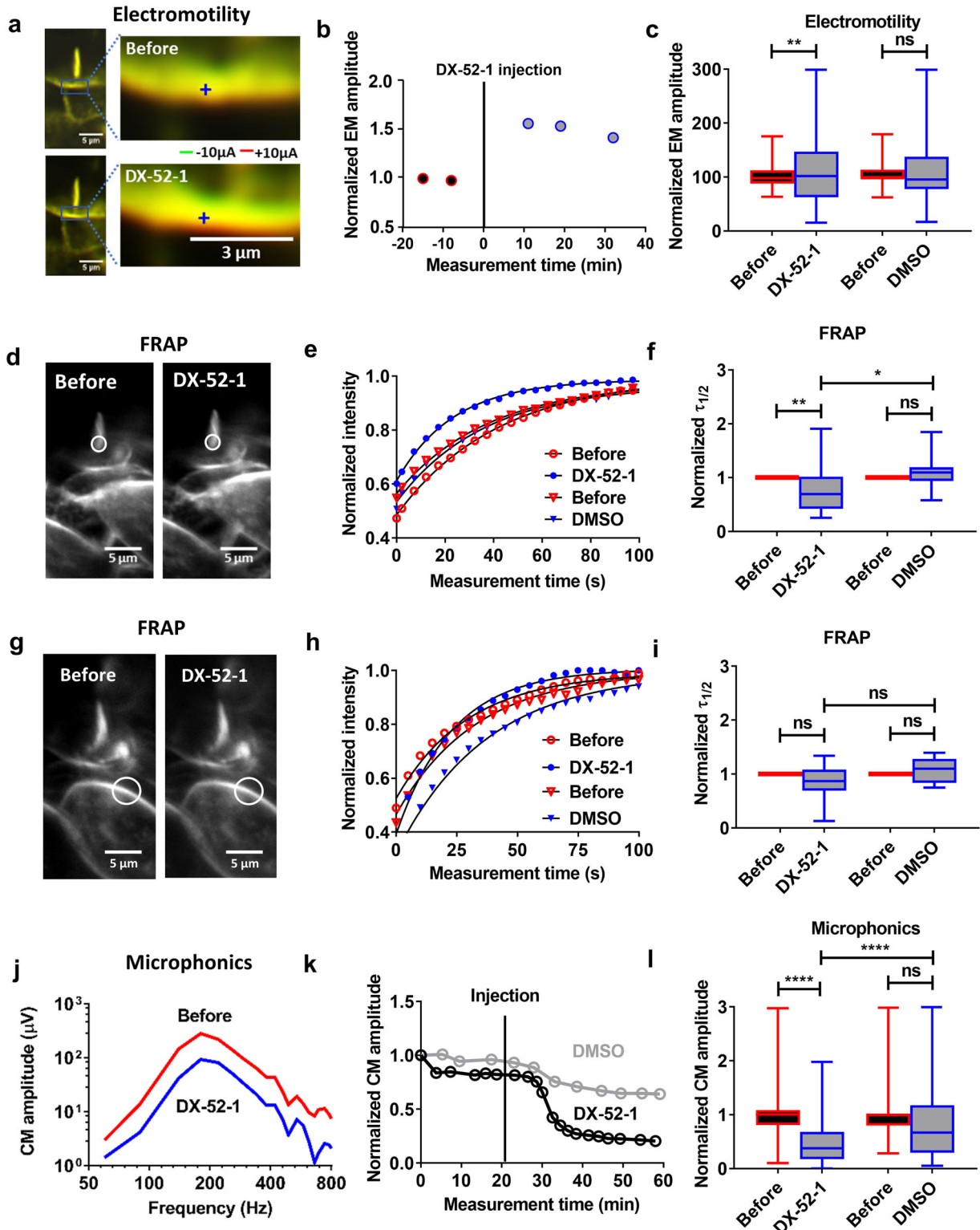

effect on mechanically sensitive channels, as demonstrated by the increase in electrically evoked motion (Fig. 4a–c), which requires currents to pass through these channels into the hair cell soma. However, the frequency of the current stimulus used for assessing electromotility is approximately 5 octaves below the best frequency of the recording location. Hence, these data suggest that radixin inhibition reduces the ability of mechanically sensitive channels to respond to stimuli at acoustic rates, but the ability to pass currents at very low frequencies into the hair cell soma is retained.

**Compound action potentials indicate loss of hearing sensitivity in vivo.** To assess the influence of radixin on hearing sensitivity in vivo, we applied 1 µl of a 1 mM DX-52-1 solution directly to the round window membrane of anesthetized guinea pigs while measuring the amplitude of the auditory nerve compound action potential (CAP). The CAP represents the summed response of auditory nerve fibers to acoustic stimulation, and is most effectively elicited by high-frequency acoustic stimuli with rapid rise time. Ten to forty minutes after the application of DX-52-1, the

**Fig. 4 DX-52-1-induced effects on OHC electromotility, lipid mobility, and microphonics. a** An OHC stereocilia bundle showing change in electrically evoked motility. Images acquired during negative current were encoded in green; images during positive current in red. **b** Time course of electromotility amplitude in an example preparation (blue circle) showing increase after DX-52-1 injection. The vertical line at time zero indicates the time of injection. Data were normalized to the average electromotility amplitude recorded before injection. **c** The average electromotility amplitude increased significantly after the DX-52-1 injection ($n = 70$) with no significant change after DMSO injection ($n = 15$). The acoustic stimulus was a 220 Hz tone at 80 dB with current stimulus of ±10 µA. **d** FRAP experiment showing lack of change in the stereocilia bundle morphology before and after DX-52-1 injection, except for a slight change in the dye intensity. **e** Normalized traces of the fluorescence intensity showing change in the membrane dynamics before and after DX-52-1 and DMSO injection in an example preparation. **f** Fitting the experimental data to single-phase exponential model showed a significantly faster recovery of bundle fluorescence with reduced $\tau_{1/2}$ after DX-52-1 injection ($n = 24$) and with no change in the diffusion time after DMSO injection ($n = 14$). **g** No change in the cell somatic membrane morphology before and after DX-52-1 injection. **h** Normalized traces of the fluorescence intensity showing no change in the membrane dynamics before and after DX-52-1 and DMSO injection. **i** Fitting the experimental data to single-phase exponential fit model showed a non-significantly faster recovery of cell membrane fluorescence with reduced $\tau_{1/2}$ after DX-52-1 ($n = 16$) injection and with no change in the diffusion time after DMSO injection ($n = 12$). Data are mean ± s.d. **j** Tuning curves for the cochlear microphonic (CM) potential before and after 1 mM DX-52-1 injection in an example preparation. **k** Example time courses of the peak amplitude of the CM potential which decreased substantially 10–15 min after DX-52-1 injection but not significantly after DMSO. The vertical line indicates the time of injection of DX-52-1 and DMSO. **l** Comparison of the average CM amplitude which reduced significantly before and after DX-52-1 injection ($n = 40$) but not after DMSO injection ($n = 11$) for experiments in panel (**k**). Data are mean ± s.d. A significant difference in the microphonic amplitude was observed between DX-52-1 and DMSO. All data sets were normalized to the data recorded before injection. ****$P < 0.0001$; ***$P < 0.001$; **$P < 0.01$; *$P < 0.05$; n.s., not significant; two-tailed paired $t$-test, two-tailed unpaired $t$-test with Welch's correction.

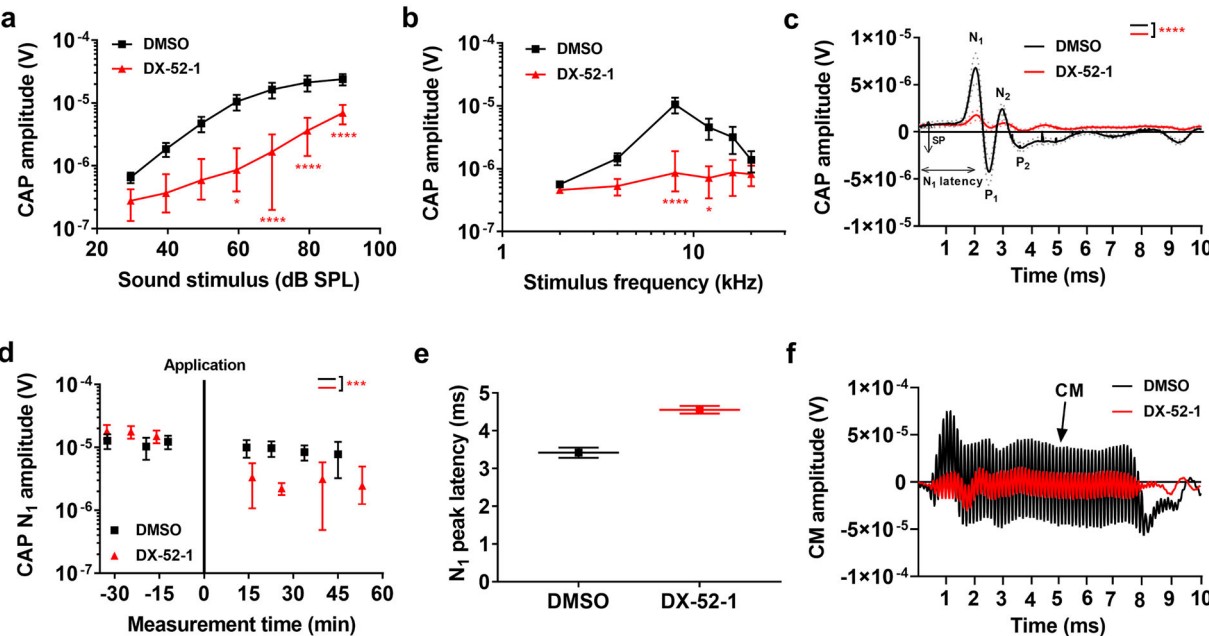

**Fig. 5 DX-52-1 results in declining hearing sensitivity, as assessed by the compound action potential of the auditory nerve. a** CAP amplitudes as a function of stimulus level at 8 kHz. **b** CAP amplitudes as a function of stimulus frequency. **c** Grand average ± s.e.m (dotted) of the CAP waveforms to 60 dB SPL stimuli shows reduction in N1 and N2 amplitudes. **d** Averaged time courses of the changes seen in N1 amplitude measured at 60 dB SPL 8-kHz stimulus following DX-52-1 application, relative to those before the application, which decreased significantly after 15–20 min of application. **e** The latency of the CAP N1 increased slightly after DX-52-1 application. **f** Representative waveforms of the CM potential, before and after 20 min of application of 1 mM DX-52-1. The stimulus was a 8-KHz tone burst at 90 dB SPL. Data information: DMSO ($n = 10$), DX-52-1 ($n = 18$). Data are mean ± s.e.m. ****$P < 0.0001$; ***$P < 0.001$; **$P < 0.01$; *$P < 0.05$; ns, not significant; two-way ANOVA coupled to the Bonferroni post hoc test, two-tailed unpaired $t$-test with Welch's correction.

CAP amplitude decreased significantly compared to control preparations where only the vehicle, perilymph with 1.8% DMSO, was applied (Fig. 5a).

Analysis of CAPs confirmed that hearing impairment was most pronounced at frequencies between 8 and 16 kHz, while smaller changes were observed at other frequencies (Fig. 5b; $n = 18$ for DX-52-1 vs. 10 controls; $P = 0.00001$; two-way ANOVA). While the overall shape of the CAP waveform remained similar after DX-52-1, there was a slight increase in the response latency (Fig. 5c, e). Figure 5d demonstrates the time course for the change in CAP N1 peak amplitude, with maximum amplitude change after about 20–30 min. As shown in Fig. 5f, DX-52-1 decreased

the amplitude of the CM potential (in Fig. 5f, the stimulus was a 90-dB SPL tone at 8 kHz; SPL, sound pressure level).

When elicited by low-level sounds, the CAP reflects the synchronous activation of neurons in cochlear regions near the place of maximum organ of Corti vibration. Worse CAP threshold would be apparent in the audiogram, and the reduced CAP amplitudes, therefore, parallel the human data, where *RDX* variants caused profound hearing loss.

**Phenylarsineoxide-induced effects on stereocilia function.** Radixin mediates interactions between the cytoskeleton and the

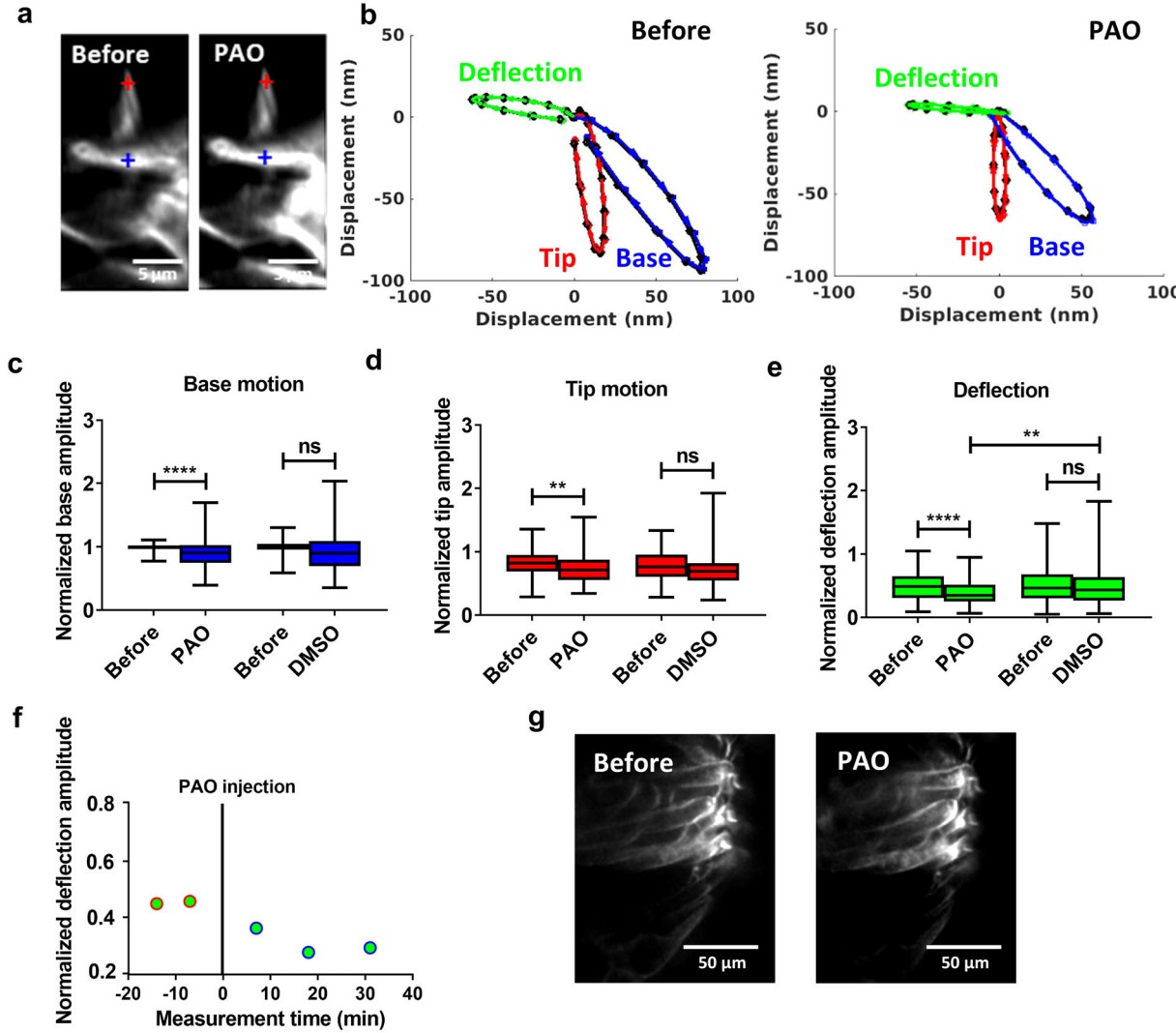

**Fig. 6 PAO-induced effects on OHC stereocilia sound-evoked motions. a** Time-resolved confocal image of an OHC stereocilia bundle showing the morphology is intact before and after the injection, except for a small change in the brightness of the fluorescent dye. **b** Representative data showing change in sound-evoked motion of the bundle tip (red) and base (blue) before and after PAO injection. The stimulus was a pure tone at 220 Hz and 80 dB sound pressure level. **c–e** Averaged change of the bundle motion at the base of outer hair cell stereocilia (blue bar), at their tip (red bar), and deflection (green bar) are shown. Significant decrease in tip and base motion resulting in a change in the bundle deflection. Data were normalized to the base trajectory amplitude recorded before the injection. Mean data from 35 (PAO) individual preparations ± s.d. **f** Example time course of deflection amplitude of outer hair cell stereocilia bundle (blue circle) showing decrease after PAO injection. The vertical line at time zero indicates the time of injection of PAO. Data were normalized to the average trajectory amplitude recorded before injection. **g** High-resolution confocal images of stereocilia in preparations treated with PAO or vehicle show no detectable morphological difference. ****$P < 0.0001$; **$P < 0.01$; *$P < 0.05$; n.s., not significant; two-tailed paired $t$-test.

cell membrane, but membrane attachment also requires the presence of PIP$_2$, the synthesis of which can be blocked by kinase inhibitors such as phenylarsineoxide (PAO)[6]. Although the rates of both fast and slow adaptation are affected by PAO[6], its indirect inhibitory effect on radixin can be used to confirm some of the DX-52-1 effects described above.

Injection of a 1-mM PAO solution into the endolymphatic space produced minor changes in brightness of outer hair cell stereocilia, but no other morphological changes were evident (Fig. 6a). As seen in the example data in Fig. 6b, the sound-evoked displacement at both the tip of the stereocilia (red trajectory) and their base (blue trajectory) decreased following PAO. This decrease led to a reduced deflection amplitude (green trajectory in Fig. 6b), even though the shapes of the motion trajectories remained similar. The change in deflection amplitude was apparent 10 – 15 min after PAO injection and the amplitude continued to be reduced over the ensuing 40 min (Fig. 6f).

Aggregated data across 35 preparations are shown in Fig. 6c–e. The decrease in motion amplitude at the base of stereocilia was significant (from 97 ± 6 nm to 86 ± 22 nm; $P = 0.00001$, two-tailed paired $t$-test; Fig. 6c) as was the change in displacement at their tips (from 80 ± 19 nm to 72 ± 22 nm; $P = 0.004$, 2-tailed paired $t$-test; Fig. 6d). The deflection amplitude decreased from 46 ± 21 nm to 37 ± 20 nm ($P = 0.00001$, two-tailed paired $t$-test; Fig. 6e). A significant difference was also found when preparations injected with PAO were compared to those injected with vehicle alone (Fig. 6e; $P = 0.004$, two-tailed unpaired $t$-test with Welch's correction). PAO caused no morphological changes to stereocilia during the time frame of the present experiments (Fig. 6g).

Considering that DX-52-1 caused an increased motion amplitude in response to electrical stimulation, we proceeded by examining the influence of PAO on electromotility. Color-coded data from an example preparation are shown in Fig. 7a. In this case, images acquired before and after PAO largely

overlapped as demonstrated by the yellow color in Fig. 7a, indicating that PAO did not change electrically evoked organ of Corti motion. There was an increase from 93 ± 36 nm to 108 ± 74 nm in the amplitude of electrically evoked motion, but this change was not significant (Fig. 7b). Across 30 preparations, there was no significant difference in the mean amplitude between preparations injected with the vehicle alone and those injected with PAO ($P = 0.20$, two-tailed paired $t$-test; Fig. 7c).

Next, we used FRAP to look for changes in the membrane lipid diffusion kinetics after PAO injection. Diffusion of di-3-ANEPPDHQ molecules within a defined region of interest (Fig. 7d) on the stereocilia was measured. In the data shown in Fig. 7e, a single-phase exponential model (black line) was fitted to the averaged fluorescence recovery curve before (red open circle) and 10 min after PAO injection (blue filled circles). The fit parameters revealed significantly faster diffusion during the

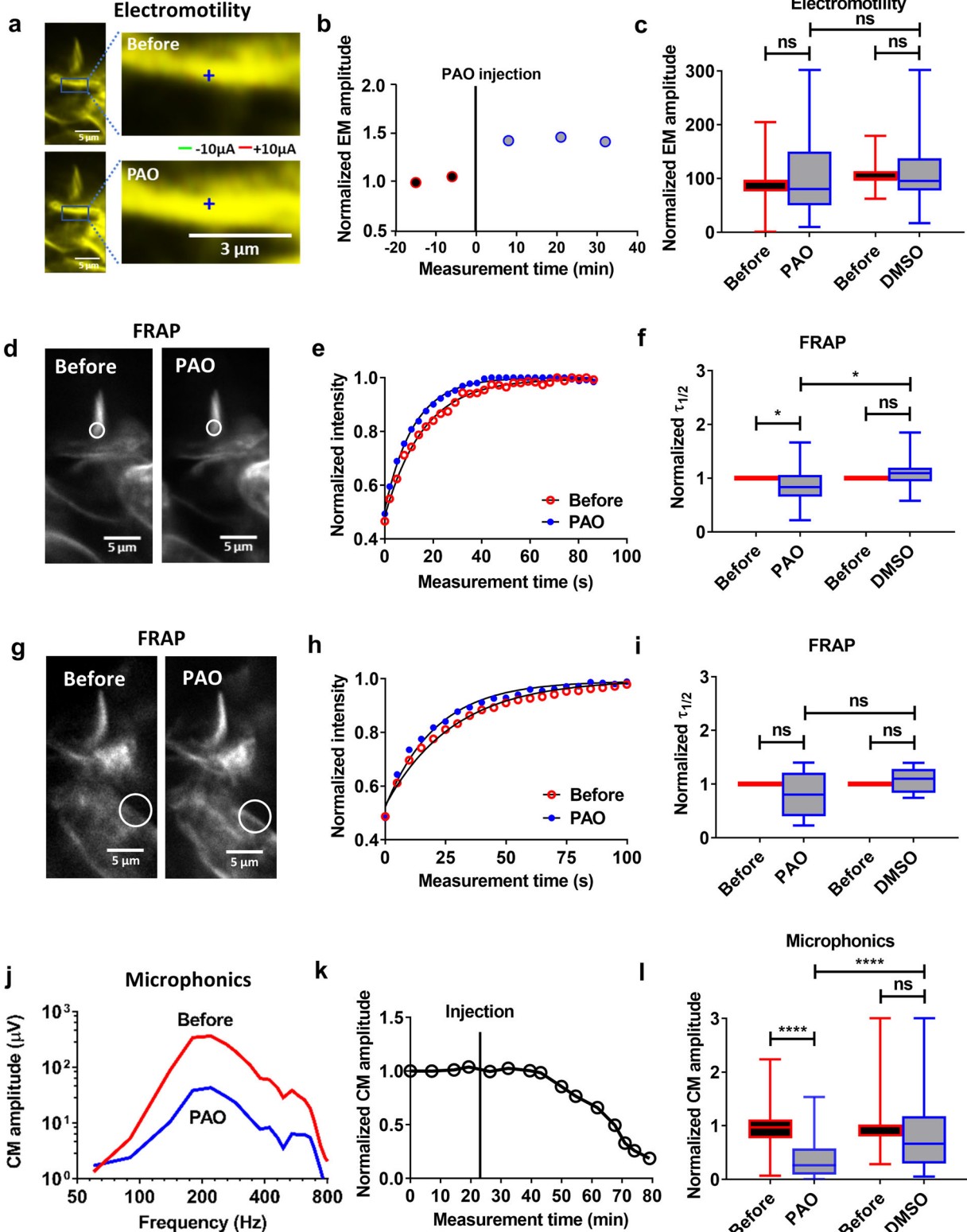

**Fig. 7 PAO-induced effects on OHC electromotility, lipid mobility, and microphonics. a** In comparison with DX-52-1 (Fig. 4a), changes in electromotility induced by PAO are small. Here, images acquired during positive and negative currents were superimposed. **b** Time course of electromotility amplitude (blue circle) in a preparation showing minor increase after PAO injection. The vertical line at time zero indicates the time of injection. Data were normalized to the average electromotility amplitude recorded before injection. **c** The average electromotility amplitude increased non-significantly after the PAO injection. Data from 30 (PAO) individual preparations. The acoustic stimulus was a 220 Hz tone at 80 dB with current stimulus of 10 μA. **d** FRAP experiment showing no change in the stereocilia bundle morphology seen before and after PAO injection. **e** Normalized traces of the fluorescence intensity during the recovery before and after PAO injection in an example preparation. **f** Fitting the experimental data to single-phase exponential fit model showed faster recovery of the bundle fluorescence with reduced $\tau_{1/2}$ after PAO injection on average for 22 preparations. **g** No change in the cell somatic membrane morphology before and after PAO injection. **h** Normalized traces of the fluorescence intensity showing no change in the membrane dynamics before and after PAO injection in an example preparation. **i** Fitting the experimental data to single-phase exponential fit model showed a non-significantly faster recovery of cell membrane fluorescence with reduced $\tau_{1/2}$ after PAO injection ($n = 9$). **j** Tuning curves for the CM potential before and after 1 mM PAO injection in an example preparation. **k** Normalized peak amplitude of the time course of the CM potential showing substantial and irreversible decrease 10–15 min after PAO injection in an example preparation. The vertical line indicates the time of injection of PAO. **l** Comparison of CM potential amplitude before and after PAO injection ($n = 33$), which reduced significantly for experiments in panel (**k**). Data are mean ± s.d. All data sets were normalized to the data recorded before the injection. ****$P < 0.0001$; **$P < 0.01$; *$P < 0.05$; n.s., not significant; two-tailed paired $t$-test.

ensuing 25–30 min (from 21 ± 16 s to 16 ± 8 s; $P = 0.04$, two-tailed paired $t$-test; $n = 22$; Fig. 7f). A significant difference in the fluorescence recovery time was also seen between preparations injected with PAO and the controls (Fig. 7f; $P = 0.03$, two-tailed unpaired $t$-test with Welch's correction). We did not find a significant change in the mobility of membrane lipids on the cell bodies of the outer hair cells after PAO injection (Fig. 7g–i, $n = 9$).

PAO injection also led to a decrease in the CM amplitude (Fig. 7j). The drop in the CM amplitude was evident within 10–15 min after the injection, and there was no recovery during the ensuing 30–40 min (Fig. 7k). On average, the CM amplitude decreased from 145 ± 96 μV to 58 ± 48 μV, measured at the peak of each tuning curve (Fig. 7l; $P = 0.00001$, two-tailed paired $t$-test; $n = 33$). A significant difference in the amplitude was seen between preparations injected with PAO and the controls (Fig. 7l; $P = 0.00001$, two-tailed unpaired $t$-test with Welch's correction). Additional data showed a minor but no significant effect on the CAP amplitudes after PAO application (Fig. 8a–f, $n = 6$).

Although neither DMSO, DX-52-1, or PAO resulted in any detectable morphological change in stereocilia (Supplementary Fig. 3a), we nevertheless performed separate experiments to evaluate stereocilia morphology. The compounds were perfused through the cochlea for 45–60 min, and the cochlea subsequently fixed in paraformaldehyde (see Methods). Fluorescence imaging of hair bundles showed no morphological difference between controls and those treated with DX-52-1 or PAO (Supplementary Fig. 3b).

The change in lipid mobility evoked by PAO and the decrease in the CM amplitude are consistent with the DX-52-1 findings; however, PAO is unspecific[29] and will affect many proteins found in stereocilia, which likely explains why the effects on sound-evoked motion and on electromotility differ from those of DX-52-1.

## Discussion
This study shows that radixin allows stereocilia to generate electrical potentials at acoustic rates, making radixin necessary for proper cochlear function. The effects of radixin inhibition, which are summarized in Fig. 9, are not due to a change in the stimulation of the sensory cells, since stereocilia deflections in outer hair cells showed a minor increase upon blocking radixin (Fig. 3). Similarly, the decrease in the electrical potentials produced by the sensory cells during acoustic stimulation is not due to inhibition of electromotility (which increased). Since electromotility requires currents to pass through MET channels, it is clear that these channels work normally at very low frequencies even after radixin inhibition, but fail to work properly at acoustic rates (the electrical stimulus was at 5 Hz, approximately 5 octaves below the best frequency of the recording location).

The FRAP experiments suggest that this pattern of functional deficits is a result of a loss of membrane–cytoskeleton interactions.

When these interactions are reduced, mechanically sensitive channels will be less firmly connected to the cytoskeleton, which influences their gating[30]. We propose that this causes an inefficient delivery of rapid stimuli to mechanically sensitive channels, which decreases the amplitude of the CM potential.

Previous studies showed that hair cell stereocilia contain high levels of radixin[1,4,5]. Studies have also demonstrated radixin labeling at the junctions between the supporting cells and the hair cells[31], but this was not evident in our experiments and consistent labeling was not found in either neurons or in the cell bodies of the sensory cells. These results suggest that radixin inhibition primarily affects stereocilia function. This view is supported by findings from radixin knockout mice, which show degeneration of stereocilia after the onset of hearing, but an otherwise normal organ of Corti structure[1]. It is possible that upregulation of ezrin, a protein closely related to radixin, ensures normal early development of stereocilia but this compensation mechanism subsequently fails. Hence, it is clear that radixin is critical during the final phases of stereocilia development, but it continues to be expressed at high levels all through the life of the animal[1] suggesting an important physiological role that has remained obscure.

Membrane-associated proteins such as radixin are often regulated by membrane lipids. Radixin is activated only after positive regulation, which requires sequential binding of PIP$_2$ and phosphorylation of threonine 564[32]. In hair cells, radixin is concentrated towards the stereocilia base, where they insert into the cuticular plate. This taper region is a site of mechanical stress during sound-evoked deflection[33]. Based on the findings of the present study, we propose that radixin, in addition to its role for channel function, contributes to the regulation of stereocilia stiffness by linking the cytoskeleton more tightly to the membrane inside this high-stress region. Findings evident after the inhibition of radixin and consistent with this hypothesis include the increased lipid mobility (Fig. 4d–f), larger electrically evoked motility (Fig. 4a–c), and larger sound-evoked stereocilia deflections (Fig. 3b, e–g). Due to the active, nonlinear mechanisms that amplify sound-evoked motion in vivo[34,35], small changes in the mechanical properties of stereocilia can have large effects on hearing organ performance.

However, the most dramatic effect of radixin inhibition was the reduction in sound-evoked electrical potentials and in the amplitude of the CAP. This demonstrates a previously unrecognized role of radixin in maintaining the normal frequency response of the mechanoelectrical transduction current. Since we (Fig. 2) could not detect radixin expression either in cochlear neurons or at the synaptic pole of the hair cells, the reduction in the CAP amplitude is explained by an effect on the transduction process itself. This finding is supported by the normal morphology of the organ of Corti in aged radixin knockout mice[1], and with the absence of obvious radixin expression in neurons.

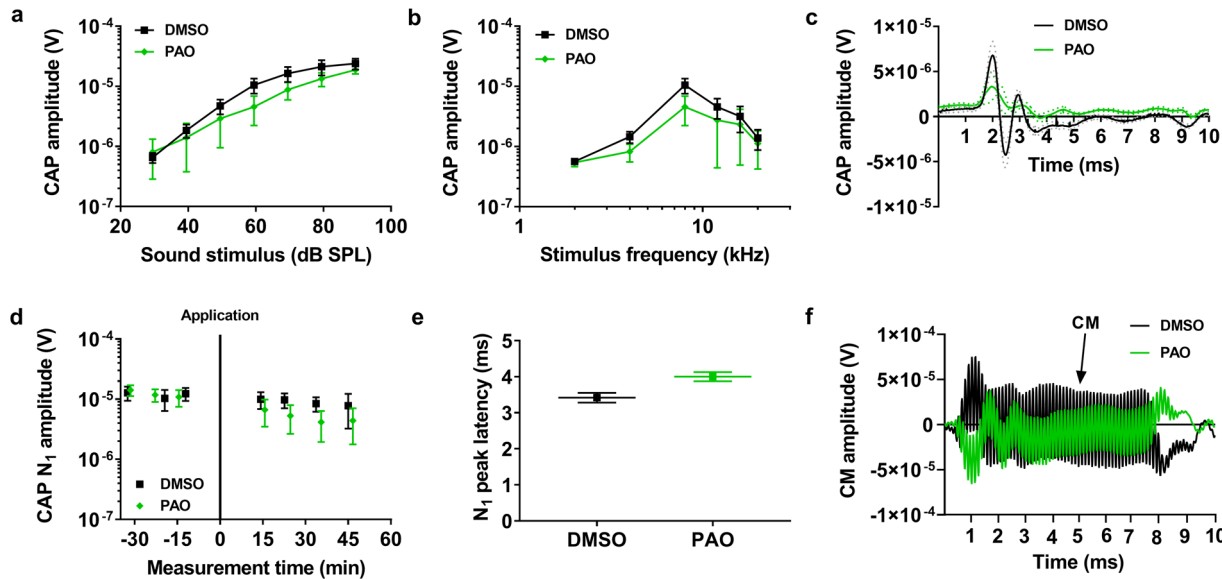

**Fig. 8 PAO effect on hearing sensitivity, assessed by the compound action potential. a** Schematic showing the CAP recordings for DMSO (black) and PAO (green) treated guinea pigs. **b** Average CAP amplitude to 60 dB SPL stimuli shows a slight reduction for the PAO animals compared to DMSO. **c** Grand averages ± s.e.m (dotted) of the CAP waveforms to 60 dB SPL stimuli shows little reduction in N1 and N2 amplitudes. **d** Averaged time courses of the changes seen in N1 amplitude measured at 60 dB SPL 8-kHz stimulus following PAO application, relative to DMSO application, which decreased non-significantly after 15–20 min of application. **e** Comparison of the CAP N1 latency after PAO application for animals in panel (**c**). **f** Representative waveforms of the CM potential, before and after 20 min of application of 1 mM PAO. The stimulus was 8-KHz tone burst at 90 dB SPL. The vertical line at time zero indicates the time of application. Data information: DMSO ($n = 10$), PAO ($n = 6$). Data are mean ± s.e.m. *$P < 0.05$; ns, not significant; two-way ANOVA coupled to the Bonferroni post hoc test, two-tailed unpaired $t$-test with Welch's correction.

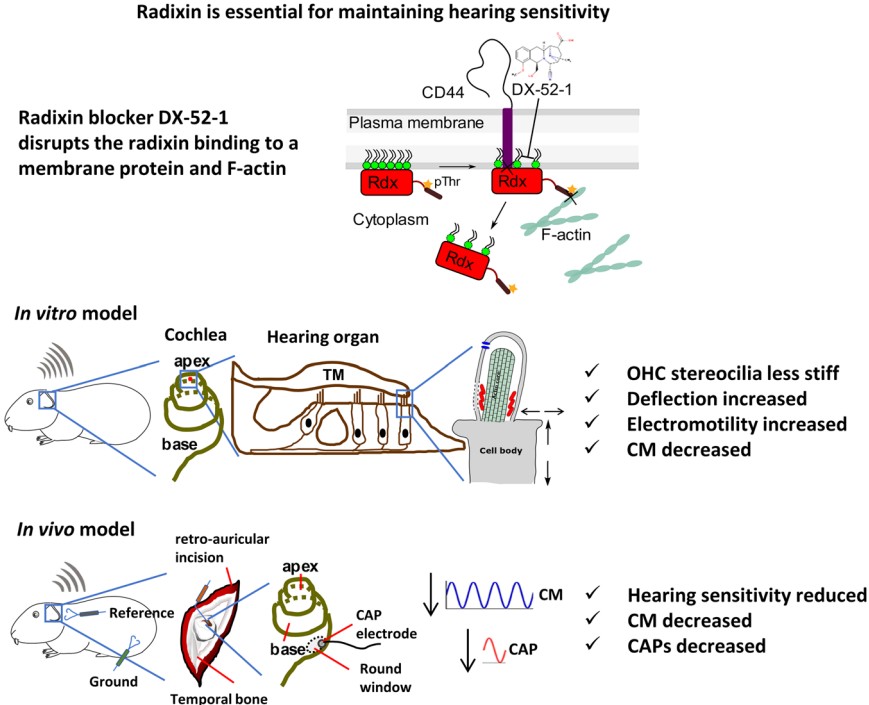

**Fig. 9 Radixin is required for maintaining the mechanical stability of stereocilia, the function of mechanically sensitive ion channels, and hearing sensitivity.** Schematic diagram of outer hair cell stereocilia with radixin-binding area showing the molecular interactions between radixin and F-actin cytoskeleton, the transmembrane protein CD 44, and the blocker mode of action. In the hearing organ of animals where the radixin blocker DX-52-1 was not applied, the animals had normal hearing and stereocilia functions. Application of the blocker results in a disruption of the link between radixin and F-actin cytoskeleton. The animal had reduced hearing sensitivity, and large effects on OHC stereocilia functions were evident.

It is interesting that two of our patients had apparently normal hearing at birth, as shown by normal newborn hearing screening results (the two siblings from pedigree 3 did not undergo neonatal hearing screening). The subsequent development of hearing loss could be due to a combination of reduced transduction currents and an inability to maintain stereocilia structure, including their stiffness, in the long term in the absence of membrane-cytoskeletal links. However, hearing loss was profound in three of our patients and moderate in one. At first sight, the removal of the start codon in exon 2 in this patient should lead to complete absence of radixin expression. Due to an in-frame start codon present in exon 3, it is however possible that a protein 11 amino acids shorter could be produced. We speculate that such a shorter protein could retain some functionality, explaining the less severe hearing loss in this patient and suggesting a clinically relevant genotype–phenotype correlation for pathogenic *RDX* variants. Moreover, in one of our families, a copy number variation contributed to the development of the hearing loss. Such a variation that is called during bioinformatics analysis can often be challenging to identify using conventional next-generation sequencing technologies.

The development of early onset hearing loss in children passing newborn hearing screening can confound both patients and their physicians, causing diagnosis and intervention to be postponed[36–38]. The resulting delays in speech and language development may contribute to impairment of social skills and cognition[39]. No previous study has examined the effect of radixin on stereocilia function. Therefore, understanding the physiopathology of genes such as *RDX* and increasing our awareness of its contribution to this burden of delayed diagnosis could improve the care of children with hearing impairment. This is important, especially for siblings of already diagnosed patients. Therefore, in these families, if a genetic diagnosis has not been obtained, close monitoring of the siblings who have passed initial newborn hearing screening is mandatory. Importantly, the fact that hearing appeared normal early in life could mean that a time window exists in the event that therapies for restoring radixin functionality become available. Nevertheless, any potential, gene-specific, therapeutic opportunity will always be enhanced by an early and comprehensive genetic diagnosis.

## Methods

**Ethics statement.** The clinical data collection was approved by the Institutional Review board at the University of Miami (USA) and by the Comité de Ética de Investigación del Principado de Asturias (research project #75/14), Spain. A signed informed-consent form was obtained from the parents of each participant. The Regional Ethics Board in Linköping approved all animal experiments (DNR 16-14) and animal care was under the supervision of the Unit for Laboratory Animal Science at Linköping University.

**Clinical study.** Patients I and II (Fig. 1) were evaluated according to standard newborn hearing screening protocols using otoacoustic emissions and/or auditory-evoked potentials. Later, patients I and II were re-evaluated because of a suspicion of hearing loss. Objective measures of hearing were used to establish their audiograms. In patient III, sensorineural hearing loss was diagnosed via standard audiometry in a sound-proof room according to current clinical standards as recommended by the International Standards Organization (ISO8253-1). Routine pure-tone audiometry was performed with age-appropriate methods to determine hearing thresholds at frequencies 0.25, 0.5, 1, 2, 4, 6, and 8 kHz. Severity of hearing loss was determined from pure-tone averages calculated at 0.5, 1.0, 2.0, and 4 kHz. Transient-evoked otoacoustic emissions were also tested. DNA was isolated from whole blood of the probands and subjected to OTOgenetics targeted gene enrichment (patients I and II) in the form of a panel of 199 hearing loss-associated genes or exome enrichment (patient III)[14,40]. Briefly, the OTOgenetics library preparation followed the SureSelectXT protocol (Agilent). Sequencing was performed using a NextSeq500 sequencer (Illumina) using manufacturer's specifications. An optimized diagnostic pipeline allowed the identification of single nucleotide variants/indels and copy number variations. Exome enrichment followed the SureSelect Human All Exon 50 Mb kit (Agilent) protocols. Sequencing was carried out using the HiSeq 2000 instrument (Illumina). An in house bioinformatics pipeline was used for variant and copy number variation calling. Validation and segregation testing of the variants was performed.

**Animal and experimental model details.** Young mature Dunkin-Hartley guinea pigs of both sexes (250–450 g; 5–6 weeks old) were used for all experiments. Prior to decapitation all animals were tested for the Preyer reflex and then anesthetized with 18–24 mg of sodium pentobarbital intraperitoneally, according to their body weight. The left temporal bone was excised and attached to a custom-built holder. The holder allowed immersion of the cochlea and the middle ear in oxygenated (95% $O_2$, 5% $CO_2$) cell culture medium (Minimum Essential Medium with Earle's balanced salts, SH30244.FS Nordic Biolabs). The bone of the bulla was gently removed to expose the middle ear and the basal turn of the cochlea, including the round window niche. Thereafter, a small triangular or trapezoidal opening was made at the apex using a #11 scalpel blade and a hole of 0.6 mm diameter was drilled at the base of the cochlea using a straight point shaped pin. These openings allowed continuous perfusion of oxygenated tissue culture medium through an external syringe tube connected to the basal hole with a plastic microtube. Sound stimulation occurred through a calibrated loudspeaker connected to the chamber with a plastic tube. Because of the immersion of the middle ear and the opening at the apex, the effective sound pressure level was reduced by ~20 dB. The values given throughout the text are corrected for this attenuation. The whole preparation was maintained at room temperature (22–24 °C). The apical opening allowed confocal imaging of the hearing organ and permitted insertion of a double-barrel glass microelectrode filled with artificial endolymph-like solution (1.3 mM NaCl, 31 mM $KHCO_3$, 23 μM $CaCl_2$, 128.3 mM KCl, pH 7.4, and 300 mOsmol/kg adjusted with sucrose) into the scala media through the Reissner's membrane. This special electrode with septum is used for CM recordings, electrical stimulation, endocochlear potential recordings, bundle membrane staining, and delivery of pharmacological substances, as specified.

**Reagents.** The following stock solutions were prepared and further diluted in artificial endolymph to the desired concentration. Di-3-ANEPPDHQ (D36801 ThermoFisher Scientific): 4 mM in pure DMSO diluted 100 times for use. Quinocarmycin analog DX-52-1 (a kind gift from the US National Cancer Institute, 96251-59-1): 22 mM in 50% DMSO and phosphate-buffered saline (PBS) diluted to 1 mM for use. Note that the effective concentration in the endolymph is lower than 1 mM because the agent is diluted in the scala media fluids upon injection. Previous estimates suggest a 10× dilution factor[16]. Phenylarsine Oxide (P3075-1G Sigma Aldrich): 45 mM in pure DMSO diluted to 1 mM for use.

**Confocal imaging.** Samples were imaged with an upright laser scanning confocal microscope (Zeiss LSM 780 Axio Imager) controlled with the ZEN 2012 software. Outer hair cell bundle displacement movements were acquired with a 40×, 0.80 numerical aperture water immersion objective lens (Zeiss Achroplan or Nikon CFI Apo lens); immunofluorescence imaging was made with a 100× oil immersion, 1.40 NA objective (Zeiss Plan-Apochromat). Images were processed in ImageJ 1.50i software, Imaris 9.2, ZEN 2012 and Matlab (R2017b, the Mathworks, Natick, MA, USA) and schemes drawn in Inkscape 0.92.3.

**Electrophysiological recordings.** Hair bundles were labeled with the membrane dye di-3-ANEPPDHQ, which was dissolved in endolymph solution and delivered by electrophoresis. This protocol ensured minimal dye release into the scala media and produced strong labeling of stereocilia while preserving the barrier function of Reissner's membrane. Double-barrel microelectrodes with an outer diameter of 1.5 mm were pulled with a standard electrode puller and beveled at 20° to a final resistance of ~4–6 MΩ. The electrodes were mounted in a manual micromanipulator at an angle of 30° and positioned in the apical opening. Reissner's membrane was penetrated using a hydraulic stepping motor. Current injections were performed with a linear stimulus isolator (A395, World Precision Instruments) sending positive steady-state currents of up to +14 μA. These currents restored the normal potential around the hair bundles, leading to an increase in the currents through the MET channel, and in the force produced by the hair cells. The endocochlear potential upon penetration of Reissner's membrane was 25–30 mV. CM potentials were measured with an IX1 amplifier (Dagan Instruments) and digitized with a 24-bit A/D board (NI USB-4431, National Instruments) at 10 kHz, using custom Labview software. Tuning curves were recorded in response to a series of tone bursts at 60 dB SPL ranging from 60 to 820 Hz. The rise and fall time was 1 ms, using a Hanning window. The sampled signals were Fourier-transformed and the peak amplitude plotted as a function of stimulus frequency. Before applying drugs, tuning curve measurements were repeated every 5 min for 15–20 min to verify that the response was stable. We thereafter proceeded with the measurements described below.

**Time-resolved confocal imaging.** To measure sound-evoked bundle motion, the hearing organ is stained with 1 μl of dye di-3-ANEPPDHQ added in the perfusion tube. Subsequently, the sensory hair cell bundles were stained with di-3-ANEPPDHQ dissolved in the electrode solution and delivered to the hair bundles iontophoretically with a current stimulus of 3–5 μA. The preparation was stimulated acoustically near the bundles' best frequency (180–220 Hz). The best frequency was selected from the highest peak of the tuning curve of the CM recordings. Image acquisition triggered both the acoustical and electrical stimulus. A series of 37 images was acquired; each series requiring ~40 s for combined sound

and current stimulus. Custom Labview software ensured that every pixel in the image series had a known phase both of the acoustic and electrical stimuli. To obtain images free from motion artifacts, the software tracked the temporal relation between the pixels and the sound stimulus. Image sequences free from motion artifacts were then reconstructed using a Fourier series approach[19,20], to generate a sequence of 12 images at equally spaced phases of the sine wave. Images for positive and negative current stimulation were also reconstructed at 12 equally spaced phases. These image sequences were low-pass filtered and subjected to optical flow analysis[20]. To improve the signal-to-noise ratio, trajectories for all pixels in 3 × 3 or 5 × 5 region were averaged. For combined sound and electrical stimuli, current injection is switched directly from positive to negative at 5 Hz to avoid charge build-up in the scala media. When imaging bundle movements, each experiment began by acquiring a baseline of 3–5 sets of images during a 20–30 min time period. Once a stable response was verified, DX-52-1, PAO, or vehicle were applied.

**Blocker injection**. For experiments in which blockers (DX-52-1 or PAO) were injected into the endolymphatic space through the double-barrel microelectrode, one barrel of the electrode was filled with the dye di-3-ANEPPDHQ dissolved in endolymph and the other contained the blocker dissolved in endolymph. Pipettes had 1.5–3 μm tip diameter and were positioned 50–70 μm from the hair bundles The blocker was pressure-injected by a 2 pound-per-square inch pressure pulse lasting for 10 s. To verify the injection, a time series of confocal images, 60–100 s in length was acquired during each injection. CM potentials were recorded before and at 5-min intervals after the injection. Sets of confocal images of hair bundle displacement were recorded before (2 sets) and after injection (3–4 sets), and continued every 5 min for the next 30–40 min of the experiment time at a stimulus level of 80 dB SPL, 10 μA at 220 Hz best frequency. The argon laser line at 488 nm and matching beamsplitter was used. To avoid bleaching, the laser power was set to the minimum value consistent with an acceptable signal-to-noise ratio.

**Fluorescence recovery after photobleaching (FRAP)**. FRAP was performed using Zeiss software, by outlining a region of interest on the stained stereocilia membrane. Following an acquisition of a series of 10 baseline images, a 2-μm spot on the stereocilia membrane was photobleached by focusing the laser at a maximum power into the region of interest[41]. The recovery of fluorescence was tracked by acquiring a series of 30 images at 1- or 3-s intervals over a time of 100–140 s. The images were 256 × 512 pixels, 12-bit pixel depth, with an integration time of 6.30 μs per pixel, and a pinhole of 1.50 Airy units. Confocal images were obtained before and 10 and 20 min apart after the blocker injection. Statistical analysis was performed by fitting the experimental data to a one-phase decay model.

**Compound action potentials (CAPs)**. To record CAPs, animals were anaesthetized with an initial dose of an intra-muscular injection of Xylazine (10 mg/kg) and Ketamine (40 mg/kg). Three to four minutes after the animal was adequately anesthetized, the surgical site of the left bulla was shaved and the animal placed on a thermostatically controlled heating blanket to maintain a core body temperature of 38 °C. Bupivacain (0.2 mg/kg), a long-acting local anaesthetic, was administered near the surgical site before skin incision. A retroauricular incision was made in order to reach the temporal bone. Muscle and other soft tissues were dissected, and the posterio-lateral part of the auditory bulla was opened to access the round window niche. A thin Teflon-insulated Ag/AgCl silver ball recording electrode was placed in close contact with the round window membrane. The electrode wire was fixed to the temporal bone with dental cement to ensure that the position of the recording electrode remained stable throughout the experiment. The animal was then placed inside a sound-proof recording booth where an Ag/AgCl electrode was inserted subcutaneously at the vertex of the skull. Cochlear CAPs were recorded sequentially from the left ear of the animal. Standardized input–output functions were generated by varying the intensity of stimulus (90 dB, 80 dB, 70 dB, 60 dB, 50 dB, 40 dB, 30 dB SPL in steps at 6 different frequencies 2 kHz, 4 kHz, 8 kHz, 12 kHz, 16 kHz, and 20 kHz). The recorded evoked-CAP signal was then filtered (high-pass frequency 3–5 Hz, low-pass frequency 3–5 kHz), amplified at a gain of 10,000 and stored for offline analysis. The responses to 200 repetitions of each stimulus were averaged with a sampling rate of 100 kHz. Three sets of recordings were obtained before the blocker application, and recordings were repeated at 5-min intervals for the next 40 min after application of the blocker. Blockers were dissolved in artificial perilymph (137 mM NaCl, 5 mM KCl, 2 mM CaCl₂, 1 mM MgCl₂, 5 mM D-glucose, 5 mM HEPES, pH 7.4, 300 mOsmol/kg) and introduced on the round window membrane. All recording softwares were written in LabVIEW.

**Surface preparation, immunofluorescence staining, and imaging**. Whole-mount preparations of the guinea pig organ of Corti were obtained as follows. Temporal bones were removed, the bony bulla was opened to visualize the cochlea, and two small holes were made in the round window and at the apex. The openings allowed perfusion of the sensory epithelium with 4% paraformaldehyde in PBS solution. In addition, in some experiments, we used these openings to perfuse DX-52-1, PAO, or DMSO for 1 h 30 min at room temperature to allow for high-resolution imaging of stereocilia after drug application (4% paraformaldehyde was perfused afterwards in these cases).The sensory epithelium was exposed by carefully removing the cochlear bone, the spiral ligament, and the tectorial membrane.

After washing the samples in PBS, they were permeabilized using 0.3% Triton X-100 soaked in 3% bovine serum albumin (0332-25G, VWR), dissolved in PBS for 10 min at room temperature followed with one-time wash with PBS for 5 min. Permeabilization was followed with blocking step by incubating the samples for 2 h in PBS containing 3% normal goat serum (NGS, 927503, BioLegend) and 3% BSA and then stained overnight at 4 °C with the primary monoclonal antibody (mouse anti-radixin, ABNOH0005962-M06, Abnova) at a dilution of 1:500. This antibody was shown to be specific for radixin by Shin et al.[5].

Samples were then washed three times with PBS for 10 min each, followed by a 2 h incubation with a mixture of the secondary antibody (goat anti-mouse Alexa Fluor® 488–conjugated IgG, ab150113 Abcam) and Alexa Fluor 568 conjugated Phalloidin (A12380, ThermoFisher Scientific) at a dilution of 1:500. The antibody solutions were prepared in blocking solution. After three washes with PBS for 20 min each, sections were readied for surface preparations. Sections of organ of Corti starting from apex to base were carefully dissected and mounted on glass slides with mounting media Fluorosave reagent (345789, Calbiochem). The slides were sealed and allowed to rest for ~2 h before proceeding with imaging. Confocal images of the mounted sections were obtained in two track channel mode with excitation at 488 nm for Alexa Fluor 488 fluorescence and at 561 nm for Alexa Fluor 568 fluorescence. Z-stacks were acquired at 12-bit pixel depth, 512 × 512 pixels, with an integration time of 6.30 μs per pixel, pinhole of 1.0 Airy units and a spacing 1.0 or 3.0 μm per slice with 20 slices up to 10 μm in total depth.

**Fluorescence intensity quantification**. Identical experimental settings and analyses were used for quantifying both radixin and phalloidin immunofluorescence. Maximum projections of confocal z-stacks were acquired and used for analysis. Organ of Corti sections were fixed, immunostained, mounted, and imaged. For background subtraction, fluorescence intensity from randomly chosen areas per preparation, lacking specific signal, were averaged and subtracted from the respective images. Hair bundles were outlined manually in ImageJ, and the average fluorescence intensity was calculated for each individual hair bundle. Individual fluorescence intensity values of a given experiment were normalized to the global average of the corresponding preparations.

**Statistics and reproducibility**. All experiments were repeated multiple times; the number of individual measurements and the number of preparations are included in the main body of the text and in the figure legends. Analyses were performed in Matlab and the statistical significance was assessed with Prism 8 (GraphPad Software, San Diego, CA, USA). Plots were generated in the Matlab and Prism softwares. A Kolmogorov-Smirnov test verified that the data were normally distributed. Differences were analyzed with Student's paired or unpaired t-test or two-way ANOVA as appropriate and were considered significant if $P < 0.05$. Details of the statistical tests used in each case are given in the text. Data expressed as mean ± s.e.m. or s.d. as indicated.

## Data availability
Sequencing data from Case 1 and Case 2 are available at the Sequence Read Archive (SRA) from the NCBI, under BioProject accession number PRJNA672676. Sequencing data from Case 3 is available at the SRA under Run ID SRR12903783. The variants have been submitted to ClinVar under the accession IDs SCV001441573 and SCV001441574. Supplementary Dataset 1 gives all numerical data on all experimental replicates for figures in the article and figures in supplementary information. Supplementary Dataset 2 gives information on the variants which are submitted to a clinical variant repository, called ClinVar. Any other source data for figures and supplementary figures are available on request from the corresponding authors.

## Code availability
The computer code for data analysis and acquisition are available from the corresponding authors upon reasonable request.

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

## Acknowledgements

This work was supported by grants from the Swedish Research Council (2018-02692 and 2017-06092), the Torsten Söderberg foundation, AFA Försäkrings AB (170069), and the County Council of Östergötland (all to A.F.), the Fundación María Cristina Masaveu Peterson (to J.C. and R.C.) as well as the US National Institutes of Health (R01DC009645 to M.T.). We thank Anna Montell Magnusson for constructive criticism on an earlier version of the manuscript, and Georg Kuhn and Åsa Sandelius for the kind gift of DX-52-1 used in initial experiments.

## Author contributions

A.F. Conceived the project, drafted and reviewed the manuscript. S.P. Designed and conceived the project, contributed to the experimental methodology, performed and designed the experiments, acquired and collected the data, analyzed and interpreted the experimental data, drafted and reviewed the manuscript. B.V. Contributed in drafting the manuscript and assisted in collaboration. G.B., M.C., R.G.-A., A.Fo., C.D.-P., M.D., J.C., R.C., A.S., and M.T.: Performed genetic and/or clinical analyses of the patients. All authors commented on the manuscript.

## Funding

## Competing interests

The authors declare no competing interests.

## Additional information

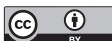

