## [Peer Review File · Communications Biology]

Reviewers' comments:

Reviewer #1 (Remarks to the Author):

This paper describes the role of radixin in moderating the translation of sound stimuli to electrical signals in the cochlea, identifying the effects of pharmacological blocking of the protein on stereocilia deflection, electromotility, the cochlear microphonic and compound action potentials. It also links these changes to the phenotypes of radixin mutations in human patients. The findings are interesting and novel, in particular the characterisation of radixin's role in modulating stereocilia stiffness; the factors underlying stereocilia movement is an important topic in auditory biophysics and is not well understood. This paper will likely be of interest to those in the auditory field, however the paper requires improvement to make the core claims convincing and to improve the clarity of the evidence presented.

Major Comments

The evidence in the paper shows that blocking radixin causes greater stereocilia deflection and electromotility, but simultaneously a reduction in the cochlear microphonic and compound action potential. The reasons for these differing responses are not sufficiently explored or discussed. The authors mention that the changes to CM and CAP are not due to inhibition of MET currents or electromotility, and suggest that it may be due to an overall change in mechanical performance, but these ideas should be explored further, as they are central to the effects of radixin inhibition on hearing performance.

Related to this point, the authors state that there is no evidence of radixin expression in cochlear neurons. I assume by this they mean the terminals contacting the IHCs, as neither the immunohistochemistry as presented, or the reference they provide (Khan et. al.) seem to show cochlear neurons beyond the terminals. I would be interested to know if the DX-52-1 inhibitor can pass the reticular lamina to reach the neurons, as ERM proteins are implicated as structural constituents of developing and mature nodes of Ranvier.

It would also be helpful for the authors to provide data on the effects of the DX-52-1 inhibitor on IHC stereocilia, particularly in light of the effects on CAP. Although OHC and IHC stereocilia are similar, they do have morphological differences that may be relevant to changes in stereocilia stiffness. Although I recognise there may be confounding effects from the changes to electromotility and the cochlear amplifier, I feel an assessment of IHC movement would strengthen the paper.

I am not sure of the usefulness of the results of blocking PIP2 synthesis in supporting the overall conclusions of the paper. Although I can see the rationale, as the authors themselves say the PAO inhibitor is non-specific, and in addition PIP2 has several other roles in the stereocilia and organ of Corti, not least an involvement in the function of the MET channel (see for example Effertz et al J. Neurosci 2017 doi: 10.1523/JNEUROSCI.1351-17.2017). These experiments need either additional data to back up the conclusion that these results are due to the interaction between radixin and PIP2, or the use of a more specific pharmacological or genetic intervention.

The general layout of the paper figures is confusing and difficult to understand for the reader. The large multipanel figures (particularly figs. 3 and 5) are difficult to follow, and contain information that I do not think needs to be presented graphically and is well explained in the text (for example the graphics showing sites of action for the inhibitors). The paper would be greatly improved by streamlining and where necessary dividing these figures so they present the individual ideas more clearly, this would improve the presentation of the paper's central interesting findings.

Minor Comments

Statistics: A justification for the use of parametric tests is required. Also, actual P values should be given and data points overlaid on bar charts. This will make the data easier to interpret for the reader.

Line 332, could the authors provide more detail in how the reduction in CAP parallels profound hearing loss? This would be helpful for general readers.

Spelling and grammar, in particular the use of tenses, needs significant attention. There are multiple examples throughout the text.

Reviewer #2 (Remarks to the Author):

The manuscript by Prasad et al., entitled "Radixin modulates stereocilia function and contributes to cochlear amplification" focuses on the human recessive deafness DFNB24. Of interest, DFNB24 babies, which carry recessive radixin (RDX) mutations are born with normal hearing, but their hearing rapidly deteriorates to profound deafness within the first year of life. Prasad et al., examine the guinea pig as model for human deafness. They use a pharmacological inhibitor of RDX to characterize acute disruption of the molecule and the consequences on hearing. Their data suggest that RDX couples the actin cytoskeleton to the membrane of outer hair cell stereocilia. The manuscript presents a thought-provoking hypothesis and a series of interesting experiments. I have several concerns listed below, attention to which would tighten up the manuscript and boost my enthusiasm for this work.

Major concerns

1) There are no control experiments validating the antibody. Hair bundles are notoriously sticky, which has led to numerous false positives. The authors go to considerable effort to demonstrate that the RDX signal is not bleed through from another channel (phalloidin). I am convinced by these data. However, whether the RDX-antibody localization is real is not clear. How can the reader be sure the staining pattern shown in Figure 2, reflects the actual localization pattern of RDX in these animals? RDX seems to be present in both outer and inner hair cells. But the manuscript focuses mostly on outers. Is there a similar function in inners?

2) The compound DX-52-1 is poorly described. The authors present this compound as if its actions are specific and its only effect is to disrupt the interaction between RDX and actin. If true, this assertion needs better documentation, either with their own data or more complete referencing to the literature. The DX-52-1 experiments are a central component of the manuscript and without a solid foundation upon which to base these experiments, the entire manuscript is called into question. The DMSO experiments provide useful controls, but they do not address whether the action of DX-52-1 is specific.

3) For the physiology presented in figure 4, it would be useful to see data not just from the DMSO control, but also from an unperfused control animals. In addition, since the authors argue that RDX inhibition affects MET channel activity, a positive control, introduction of an MET blocker would boost confidence in their drug delivery system and their ability measure the consequences of changes to MET function.

4) I do not follow the author's logic in the discussion. They argue that RDX contributes to stereocilia stiffness and suggest that inhibition of RDX leads to larger stereocilia deflections (lines 418-422). This part makes sense. Larger deflections could be expected to lead to greater MET activation and larger receptor potentials. Instead they suggest there is a "decrease in electrical potentials produced by hair cells" (line 395-396). This doesn't make sense. If indeed the case, a decrease in the outer hair cell receptor potential should decrease somatic electromotility, a voltage-dependent process. However, they argue there is greater motility when RDX is disrupted

with DX-52-1. This series of events defies conventional understanding and requires better explanation with a clearly mapped out logical flow of ideas.

5) Given concern #4 above, it is not clear how the authors jump to the conclusion stated in the title, that Radixin contributes to cochlear amplification, when they argue there is greater motility when RDX is disrupted. At a minimum, this seems an overstatement and should be toned down.

6) Lastly, the writing needs attention throughout, the abstract, in particular, is not clear. And here again, the conclusion that radixin contributes to cochlear amplification is overstated and needs to be toned down

Reviewer #3 (Remarks to the Author):

Radixin modulates stereocilia function and contributes to cochlear amplification

In this paper, Prasad et al. aim at addressing the physiological function of Radixin in hair cells, reporting also some human RDX/DNFB24 cases. The human cases have early onsets and for 2/3 of them likely occurring after birth. The major work of this paper is to analyze the effects of the pharmacological inactivation of Radixin in a "close to in vivo" mammalian model, using adult guinea pig. The authors use 1) live imaging to estimate the motion of the hair bundle of outer hair cell where Radixin localizes at the stereocilia base with their antibody; 2) electric stimulation changing the electric potential in the scala media, increasing the amount of current through the MET channels, to estimate the amount of somatic amplification; 3) FRAP on the hair bundle to estimate the lipid diffusion of the stereocilia membrane via the half recovery time; 4) measure cochlear microphonics currents resulting and compound action potential at the auditory nerve, downstream of the hair cell function.

The authors found that using a drug blocking radixin function, the hair bundle motion was slightly increased, like the electromotility, while the lipid diffusion was increased and the cochlear microphonics and CAP were reduced, in accordance with a role of Radixin in providing some stiffness to the hair bundle required for its functionality. As Radixin requires PIP2 to bind to membrane, the authors explore if the depletion of PIP2 in the organ of Corti induces similar effects as the pharmacological depletion of Radixin, and for a large part they do. They conclude that Radixin is necessary for cochlear amplification likely by decreasing stereocilia stiffness.

While this original experimental model and approach can go beyond the current mouse genetic model for radixin (which show hair bundle defect after the onset of hearing) and provide novel claims important for the field, I found some critical issues in the work.

First and most importantly, all the work relies on the specificity of the drug DX-52-1 for Radixin in the organ of Corti. This drug has been elegantly shown to be specific for Radixin in MDCK cells, and to a far lesser degree to HSP70 (Kahsai, 2006). Nevertheless, the authors do not demonstrate that the drug is selectively specific for Radixin in the organ of Corti. Caution should be taken also as one report used this drug to block the activity of Moesin (Zhu, 2013). The authors could tag the drug and do pull-down and mass spec, and use it to label the guinea pig inner ear to see where it binds to, for example. Also, if the drug works as stated, binding to the membrane and the actin should be abolished, and potentially the protein could be mislocalized, which would demonstrate that at least Radixin is affected, which for now is not done and only assumed. Along these lines, it would be important to have other tools than this antibody (which specificity has not been shown or discussed) to look at Radixin expression (in situ hybridization, NGS data) to confirm or not if Radixin is expressed selectively in hair cells in the adult inner ear of mammals. The authors mentioned that as ref#2 support the absence of spiral ganglion neuron expression, their CAP latency results cannot be due to a function in these cells. However, I could not find the supporting data in the reference. Imaging at high resolution (Fluorescence and EM) the hair bundles after

DMSO and drugs would be important to insure that there is no dramatic morphological defect induced to the organ of Corti and the hair bundle.

Secondly I have some concerns with some important part of experiments;

- For the hair bundle motion, according to Fig 3C and F, the range of motion observed in control can vary a lot, and the authors had to normalize the values. What is not clear to me is how they calculated the Standard Deviation of these ratio, as it is not a straight forward thing (propagation of error) and it is crucial for the downstream statistical tests, especially because the differences are minimal.

- For the electromobility measurements, the raw data presented (Fig 3 J) I can see only a minor difference upon DX-52-1 drug, and the "Before treatments" images looks very different from the similar experiment with PAO (Fig 5H). Again, the importance of how the SD was determined is crucial for the statistical tests.

- For the FRAP data of Fig3, I don't understand how the half-recovery time can be compared between the conditions while the amount of bleached fluorescence vary so much. This could indicate that the recovering lipid pools could contribute differently between conditions. However, this seems appropriately controlled in the PAO experiment shown Fig. 5K. A good internal control could be to perform the experiment on the hair cell somatic membrane, that according to their model should not be affected.

I think it would be determinant to address these points.

Response to the reviewers

We thank the reviewers for the detailed comments and suggestions, which led to significant improvements. In the text below, reviewer comments are given in bold type, followed by our response.

Reviewer #1: This paper describes the role of radixin in moderating the translation of sound stimuli to electrical signals in the cochlea, identifying the effects of pharmacological blocking of the protein on stereocilia deflection, electromotility, the cochlear microphonic and compound action potentials. It also links these changes to the phenotypes of radixin mutations in human patients. The findings are interesting and novel, in particular the characterisation of radixin's role in modulating stereocilia stiffness; the factors underlying stereocilia movement is an important topic in auditory biophysics and is not well understood. This paper will likely be of interest to those in the auditory field, however the paper requires improvement to make the core claims convincing and to improve the clarity of the evidence presented.

We thank the reviewer for these positive comments. As detailed below, we performed additional experiments and also made substantial changes to the text and figures to make sure the claims are well founded and clearly presented.

The evidence in the paper shows that blocking radixin causes greater stereocilia deflection and electromotility, but simultaneously a reduction in the cochlear microphonic and compound action potential. The reasons for these differing responses are not sufficiently explored or discussed. The authors mention that the changes to CM and CAP are not due to inhibition of MET currents or electromotility, and suggest that it may be due to an overall change in mechanical performance, but these ideas should be explored further, as they are central to the effects of radixin inhibition on hearing performance.

The discussion now includes additional material to clarify the nature of the functional changes observed upon blocking radixin. Parts of the results section were also rewritten.

In particular, we should more clearly have pointed to the fact that the electrical stimulus is quasi-static (5 Hz, about 5 octaves below the best frequency of the recording location). The increase of electromotility may seem puzzling considering that the cochlear microphonic potentials were reduced. However, since the electrical stimulus is at a very low frequency, our interpretation is that the MET channels retain the ability to pass currents at low frequencies (quasi-static) but fail to work properly at acoustic frequencies after radixin inhibition.

Related to this point, the authors state that there is no evidence of radixin expression in cochlear neurons. I assume by this they mean the terminals contacting the IHCs, as neither the immunohistochemistry as presented, or the reference they provide (Khan et. Al.) seem to show cochlear neurons beyond the terminals. I would be interested to know if the DX-52-1 inhibitor can pass the reticular lamina to reach the neurons, as ERM proteins are implicated as structural constituents of developing and mature nodes of Ranvier.

We performed additional experiments to clarify this important issue. The starting point for these experiments is the observation that radixin block increases the membrane mobility of stereocilia (Figure 4). A similar effect is expected in neurons expressing radixin, where radixin is also thought to link the membrane with the underlying cytoskeleton. Hence, we performed FRAP experiments on the dendrites of the auditory nerve where they cross the inner sulcus region. These experiments, which are included in supplementary figure 3, show no change in membrane

mobility following DX-52-1 application in scala media. Additional FRAP experiments were performed on the cell bodies of the outer hair cells. There was no change in membrane mobility after DX-52-1 injection (Supplemental Figure 2).

The immunohistochemistry experiments reported in figure 2 showed no detectable radixin expression in either the neurons, the terminals under the inner hair cells, or the cell bodies of the outer hair cells, as stated in the original version of the manuscript and as shown in Figure 2F. Hence, we conclude that effects at the neuronal level do not explain the reduction in CAP amplitudes seen after DX-52-1.

It would also be helpful for the authors to provide data on the effects of the DX-52-1 inhibitor on IHC stereocilia, particularly in light of the effects on CAP. Although OHC and IHC stereocilia are similar, they do have morphological differences that may be relevant to changes in stereocilial stiffness. Although I recognise there may be confounding effects from the changes to electromotility and the cochlear amplifier, I feel an assessment of IHC movement would strengthen the paper.

We agree, and consequently performed a new set of experiments. However, inner hair cells are less numerous than outer hair cells and in most cases inner hair cell stereocilia do not stain as well with di-3-ANEPPDHQ as the outer hair cells. Despite a considerable effort we were only able to obtain a sample of 18 inner hair cells. In these experiments, there was no effect of DX-52-1 on sound-evoked motion. As noted by the reviewer, there are functional and morphological differences between inner and outer hair cells, which is a possible explanation for these findings. It is also possible that a minor effect on inner hair cell bundle movements is present (Supplemental Figure S1), but we lack the statistical power to detect it. Given the experimental difficulties in measuring inner hair cells bundle movements, it would be exceedingly laborious to generate a sample of inner hair cells the same size as the one we have for outer hair cells.

I am not sure of the usefulness of the results of blocking PIP2 synthesis in supporting the overall conclusions of the paper. Although I can see the rationale, as the authors themselves say the PAO inhibitor is non-specific, and in addition PIP2 has several other roles in the stereocilia and organ of Corti, not least an involvement in the function of the MET channel (see for example Effertz et al J. Neurosci 2017 doi: 10.1523/JNEUROSCI.1351-17.2017). These experiments need either additional data to back up the conclusion that these results are due to the interaction between radixin and PIP2, or the use of a more specific pharmacological or genetic intervention. Unfortunately, we are not aware of pharmacological tools that are more specific, and anatomical constraints as well as the fragility of the mouse cochlea means that transgenic mice cannot be used for imaging sound-evoked stereocilia movements. As pointed out by the reviewer, there is a rationale for these experiments, and we therefore left these results within the paper. However, given that we cannot presently see a way to make these experiments more specific, we are willing to move these results to the supplementary data if the reviewer finds it necessary.

The general layout of the paper figures is confusing and difficult to understand for the reader. The large multipanel figures (particularly figs. 3 and 5) are difficult to follow, and contain information that I do not think needs to be presented graphically and is well explained in the text (for example the graphics showing sites of action for the inhibitors). The paper would be greatly improved by streamlining and where necessary dividing these figures so they present the

individual ideas more clearly, this would improve the presentation of the paper's central interesting findings.

We appreciate the concerns of the reviewer, and in an effort to make the figures and text easier to understand, figures 3 and 5 were divided into several individual figures and the text updated.

Statistics: A justification for the use of parametric tests is required. Also, actual P values should be given and data points overlaid on bar charts. This will make the data easier to interpret for the reader.

The Kolmogorov-Smirnov test showed that the distribution of our data was not different from the normal distribution, and we therefore used parametric tests. This information was added to the methods section. Unfortunately, overlaying data points on bar graphs will make the graph look overcrowded, given the substantial number of experiments that we performed (n=70). Also, the data were normally distributed and the standard deviation and mean therefore provide a good description of the data set.

Line 332, could the authors provide more detail in how the reduction in CAP parallels profound hearing loss? This would be helpful for general readers.

We appreciate this comment and inserted additional text to highlight the relevance of these findings for the general reader.

Spelling and grammar, in particular the use of tenses, needs significant attention. There are multiple examples throughout the text.

We made efforts to improve this and hope that these changes address the problems found by the reviewer.

Reviewer #2:

The manuscript by Prasad et al., entitled "Radixin modulates stereocilia function and contributes to cochlear amplification" focuses on the human recessive deafness DFNB24. Of interest, DFNB24 babies, which carry recessive radixin (RDX) mutations are born with normal hearing, but their hearing rapidly deteriorates to profound deafness within the first year of life. Prasad et al., examine the guinea pig as model for human deafness. They use a pharmacological inhibitor of RDX to characterize acute disruption of the molecule and the consequences on hearing. Their data suggest that RDX couples the actin cytoskeleton to the membrane of outer hair cell stereocilia. The manuscript presents a thought-provoking hypothesis and a series of interesting experiments. I have several concerns listed below, attention to which would tighten up the manuscript and boost my enthusiasm for this work.

We thank the reviewer for these positive comments. In this revised version, we made efforts to tighten the text and improve the presentation of the findings.

1) There are no control experiments validating the antibody. Hair bundles are notoriously sticky, which has led to numerous false positives. The authors go to considerable effort to demonstrate that the RDX signal is not bleed through from another channel (phalloidin). I am convinced by these data. However, whether the RDX-antibody localization is real is not clear. How can the reader be sure the staining pattern shown in Figure 2, reflects the actual localization pattern of RDX in these animals?

The same antibody was used by several groups before us, including Zhao et al (J Neurosci 2012), Bahloul et al (EMBO Mol Med 2009), Persson et al (Front Cell Neurosci 2013) and Shin et al

(Nature Neurosci 2012). Shin and colleagues verified the antibody results using mass spectrometry on isolated chick vestibular hair bundles. It is well established that the antibody we use targets radixin.

Furthermore, the pattern of stereocilia labeling is identical to the one found in previous studies (stronger labeling near the base of stereocilia and tapering of the fluorescence on approaching their tip). Given that this antibody is extensively used and that our results are similar to the ones from frogs, chicks, and mice, it seems clear that the pattern we observe is real.

RDX seems to be present in both outer and inner hair cells. But the manuscript focuses mostly on outers. Is there a similar function in inners?

We agree that it is important to assess the role of RDX in both outer and inner hair cells, and therefore conducted a new set of experiments where inner hair cell bundle movements were imaged during sound stimulation. Note that this experiment is considerably more challenging than imaging outer hair cells, because there are fewer inner hair cells and their stereocilia are less prone to be stained by the membrane dye that we use. In the more limited sample of 18 inner hair cell experiments that we managed to obtain, there was no significant effect on bundle movement from DX-52-1. This may reflect a true functional difference between the cell types, but it is also possible that a small effect is present and that we lack the statistical power to detect it. However, it would be exceedingly laborious to get the same sample size as we have in the case of the outer hair cells.

2) The compound DX-52-1 is poorly described. The authors present this compound as if its actions are specific and its only effect is to disrupt the interaction between RDX and actin. If true, this assertion needs better documentation, either with their own data or more complete referencing to the literature. The DX-52-1 experiments are a central component of the manuscript and without a solid foundation upon which to base these experiments, the entire manuscript is called into question. The DMSO experiments provide useful controls, but they do not address whether the action of DX-52-1 is specific.

New material was added to more fully discuss the specificity of DX-52-1. Khsai et al (Chem Biol 2006) showed that radixin is the main target of DX-52-1. There was also some binding to HSP70, but this binding was “noncompetable and nonspecific” and had no effect on the ATPase activity of HSP70 in vitro (Khsai et al 2006).

In a subsequent report (Khsai et al J Biol Chem 2008) the binding partners of DX-52-1 were characterized in more detail. The authors note that radixin was, by far, the most intensely labeled DX-52-1 binding protein. Galectin-3 was however identified as a secondary target. This protein is present in hair cells, but galectin-3 KO mice have normal ABRs and normal acoustic startle response (see <https://www.mousephenotype.org/data/genes/MGI:96778>). Since the hearing organ is unaffected when galectin-3 is knocked out, inhibition of galectin-3 is unlikely to explain our results.

Khsai et al (Biochim Biophys Acta 2010) evaluated the binding of DX-52-1 to ezrin, moesin and radixin. In figure 8A of that paper, strong binding of DX-52-1 to radixin is evident along with weaker binding to ezrin and even less binding to moesin. The authors conclude that DX-52-1 “exhibits considerable selectivity for radixin over the other ERM proteins”. Ezrin and moesin are not detectable in the mature hearing organ (Kitajiri et al 2004).

DX-52-1 was also evaluated by Persson et al (Front Cell Neurosci 2013). At 50 nM and above there was a significant reduction in the migration of PS-NCAM+ neuroblasts, which express radixin. Cell lines not expressing radixin had normal migration, which also shows the effect is not due to general toxicity. In an in vivo model, DX-52-1 reduced the amount of phosphorylated radixin in the rostral migratory stream of the subventricular zone of the rat brain.

In summary, a highly specific effect on radixin is expected when DX-52-1 is administered to the inner ear. This specificity arises in part because of chemical specificity and in part because other potential binding partners are not present or have no functional role in the hearing organ.

3) For the physiology presented in figure 4, it would be useful to see data not just from the DMSO control, but also from an unperfused control animals.

There was no change in the CAP amplitudes when DMSO was applied. This can be seen by comparing the figure below to Figure 5 in the new version.

Fig. 1. Compound action potential for unperfused animals. (A) Schematic showing the CAP recordings for non-treated guinea pigs (n=27) at 8-kHz. (B) Average CAP amplitude shows no reduction for the unperfused animals compared to DMSO to 60 dB SPL stimuli. (C) Grand average \pm s.e.m (dotted) of the CAP waveforms shows no reduction in N1 and N2 amplitudes at 60 dB SPL 8-kHz stimulus.

In addition, since the authors argue that RDX inhibition affects MET channel activity, a positive control, introduction of an MET blocker would boost confidence in their drug delivery system and their ability measure the consequences of changes to MET function.

We used this drug delivery system for many years and are confident in its use. In Fig. 2, the MET channel blocker FM1-43 was driven into the endolymph by positive current. This resulted in a reduced amplitude of the cochlear microphonic potential (control experiments showed that the reduction was not due to the current).

Fig. 2. The transduction channel blocker FM1-43 decreases the amplitude of the cochlear microphonic potential. Squares denote data before FM1-43 application, asterisks data afterwards, and the line the noise level of the recording system. Data from Jacob et al, Biophys J 2011.

Fig. 3. Pressure-injection of a low concentration of EGTA decreased the amplitude of the cochlear microphonic potential. This change was partly reversed by pressure-injecting artificial endolymph with high calcium content. Data from Strimbu et al, PNAS 2019.

We also use pressure injection, and Fig. 3 shows

data where a low concentration of the calcium chelator EGTA was injected into the endolymph.

This resulted in reduced cochlear microphonic amplitudes, which were partially reversible upon injection of high-calcium endolymph.

Hence, we have considerable experience with drug delivery to scala media and the technique is both reliable and reproducible. The experiments reported in the present paper would not work unless the drug delivery works, since the only way to reliably stain stereocilia is to inject the dye through the pipette, directly into the endolymph.

4) I do not follow the author's logic in the discussion. They argue that RDX contributes to stereocilia stiffness and suggest that inhibition of RDX leads to larger stereocilia deflections (lines 418-422). This part makes sense. Larger deflections could be expected to lead to greater MET activation and larger receptor potentials. Instead they suggest there is a "decrease in electrical potentials produced by hair cells" (line 395-396). This doesn't make sense. If indeed the case, a decrease in the outer hair cell receptor potential should decrease somatic electromotility, a voltage-dependent process. However, they argue there is greater motility when RDX is disrupted with DX-52-1. This series of events defies conventional understanding and requires better explanation with a clearly mapped out logical flow of ideas.

We agree that this was poorly explained and have taken steps to remedy the problem.

As pointed out by the reviewer, the increased electromotility is puzzling considering that the cochlear microphonic potentials were much reduced. However, since the electrical stimulus is essentially static – 5 Hz current steps delivered to scala media – our interpretation is that the MET channels pass current at low frequencies (quasi-static) but fail to work properly at acoustic frequencies after radixin inhibition. This may be a consequence of a DX-52-1-induced loss of coupling between the membrane and the cytoskeleton (as suggested by the FRAP experiments). We note that gating of the NOMPC channel essential for *Drosophila* hearing requires connection to the cytoskeleton (Zhang et al, Cell 2015). The increase in electromotility is a consequence of decreased stereocilia stiffness.

We inserted new material in the discussion and updated the text of the results section in several places.

5) Given concern #4 above, it is not clear how the authors jump to the conclusion stated in the title, that Radixin contributes to cochlear amplification, when they argue there is greater motility when RDX is disrupted. At a minimum, this seems an overstatement and should be toned down.

We changed the title to avoid overstating our result. The reason we think this is important for cochlear amplification is the apparent failure of MET channels to pass current at physiologically relevant frequencies following RDX inhibition.

6) Lastly, the writing needs attention throughout, the abstract, in particular, is not clear. And here again, the conclusion that radixin contributes to cochlear amplification is overstated and needs to be toned down

We rewrote parts of the abstract as suggested by the reviewer.

Reviewer #3:

In this paper, Prasad et al. aim at addressing the physiological function of Radixin in hair cells, reporting also some human RDX/DNFB24 cases. The human cases have early onsets and for 2/3 of them likely occurring after birth. The major work of this paper is to analyze the effects of the pharmacological inactivation of Radixin in a “close to in vivo” mammalian model, using adult guinea pig. The authors use 1) live imaging to estimate the motion of the hair bundle of outer hair cell where Radixin localizes at the stereocilia base with their antibody; 2) electric stimulation changing the electric potential in the scala media, increasing the amount of current through the MET channels, to estimate the amount of somatic amplification; 3) FRAP on the hair bundle to estimate the lipid diffusion of the stereocilia membrane via the half recovery time; 4) measure cochlear microphonics currents resulting and compound action potential at the auditory nerve, downstream of the hair cell function. The authors found that using a drug blocking radixin function, the hair bundle motion was slightly increased, like the electromotility, while the lipid diffusion was increased and the cochlear microphonics and CAP were reduced, in accordance with a role of Radixin in providing some stiffness to the hair bundle required for its functionality. As Radixin requires PIP2 to bind to membrane, the authors explore if the depletion of PIP2 in the organ of Corti induces similar effects as the pharmacological depletion of Radixin, and for a large part they do. They conclude that Radixin is necessary for cochlear amplification likely by decreasing stereocilia stiffness. While this original experimental model and approach can go beyond the current mouse genetic model for radixin (which show hair bundle defect after the onset of hearing) and provide novel claims important for the field, I found some critical issues in the work.

We thank the reviewer for the positive comments. We of course agree that it is important to more closely examine the consequences of RDX inhibition and hope that the explanations below and the changes made to the manuscript will resolve the issues highlighted by the reviewer.

First and most importantly, all the work relies on the specificity of the drug DX-52-1 for Radixin in the organ of Corti. This drug has been elegantly shown to be specific for Radixin in MDCK cells, and to a far lesser degree to HSP70 (Kahsai, 2006). Nevertheless, the authors do not demonstrate that the drug is selectively specific for Radixin in the organ of Corti. Caution should be taken also as one report used this drug to block the activity of Moesin (Zhu, 2013).

New material was added to more fully discuss the specificity of DX-52-1. Kahsai et al (Chem Biol 2006) showed that radixin is the main target of DX-52-1. There was also some binding to HSP70, but this binding was “noncompetable and nonspecific” and had no effect on the ATPase activity of HSP70 in vitro (Kahsai et al 2006).

In a subsequent report (Kahsai et al J Biol Chem 2008) the binding partners of DX-52-1 were characterized in more detail. The authors note that radixin was, by far, the most intensely labeled DX-52-1 binding protein. Galectin-3 was however identified as a secondary target. This protein is present in hair cells, but galectin-3 KO mice have normal ABRs and normal acoustic startle response (see <https://www.mousephenotype.org/data/genes/MGI:96778>). Since the hearing organ is unaffected when galectin-3 is knocked out, inhibition of galectin-3 is very unlikely to explain our results.

Kahsai et al (Biochim Biophys Acta 2010) evaluated the binding of DX-52-1 to ezrin, moesin and radixin. In figure 8A of that paper, strong binding of DX-52-1 to radixin is evident along with weaker binding to ezrin and even less binding to moesin. The authors conclude that DX-52-1 “exhibits considerable selectivity for radixin over the other ERM proteins”. While Zhu et al used

DX-52-1 to block moesin, they also comment that the drug is much less effective at blocking moesin than it is at blocking radixin. Furthermore, ezrin and moesin are not detectable in the mature hearing organ (Kitajiri et al J Cell Biol 2004), while radixin is one of the most highly concentrated proteins within bundles (Shin et al Nature Neurosci 2012).

DX-52-1 was also evaluated by Persson et al (Front Cell Neurosci 2013). At 50 nM and above there was a significant reduction in the migration of PS-NCAM+ neuroblasts, which express radixin. Cell lines not expressing radixin had normal migration, which shows the effect is not due to general toxicity. In an in vivo model, DX-52-1 reduced the amount of phosphorylated radixin in the rostral migratory stream of the subventricular zone of the rat brain.

In summary, a highly specific effect on radixin is present when DX-52-1 is administered to the inner ear. This specificity arises in part because of chemical specificity and in part because other potential binding partners are not present or have no functional role in the hearing organ.

The authors could tag the drug and do pull-down and mass spec, and use it to label the guinea pig inner ear to see where it binds to, for example. Also, if the drug works as stated, binding to the membrane and the actin should be abolished, and potentially the protein could be mislocalized, which would demonstrate that at least Radixin is affected, which for now is not done and only assumed.

Given the rather extensive literature on DX-52-1 specificity (see above), and the fact that the secondary targets of DX-52-1 either are not detectable in the hearing organ (ezrin, moesin) or lack functional relevance (galectin-3 knockout mice have normal hearing), we do not think that the experiments proposed by the reviewer would add any new information.

We note that fluorescent analogs of DX-52-1 were reported to have reduced activity, so this strategy is not straightforward and would require new chemical synthesis to be carried out since neither DX-52-1 nor its fluorescent analogs are commercially available. Pull-down and mass-spec and different assays was used to study the specificity of DX-52-1 and its binding with radixin, but using these techniques on the organ of Corti is challenging due to the small number of hair cells in each cochlea.

Along these lines, it would be important to have other tools than this antibody (which specificity has not been shown or discussed) to look at Radixin expression (in situ hybridization, NGS data) to confirm or not if Radixin is expressed selectively in hair cells in the adult inner ear of mammals.

The same antibody was used by several groups before us, including Zhao et al (J Neurosci 2012), Bahloul et al (EMBO Mol Med 2009), Persson et al (Front Cell Neurosci 2013) and Shin et al (Nature Neurosci 2012). Shin and colleagues verified the antibody results using mass spectrometry on isolated chick vestibular hair bundles. It is well established that the antibody we use targets radixin.

Since previous work shows radixin is present at high levels in mammalian stereocilia and our immunohistochemical data are consistent with these findings, we do not think that either in situ hybridization or NGS approaches would reveal anything that is not already known. We note that a consistent labeling pattern was found in chicks, frogs, and mice, and the pattern we show is consistent with these previous results.

To further show that radixin is present in mammalian hair cells, one of us (Barbara Vona) studied the expression of *RDX* in the mouse cochlea, using a database with single-cell RNA-seq data from inner hair cells, outer hair cells, and Deiter cells (<https://morlscrnaseq.org/>; PMID: 30865901). Expression was verified in all three cell types. However, most of the available immunohistochemistry data show little expression of radixin in supporting cells, and very high levels in hair cell stereocilia, so there may be differences between the RNA and the protein levels that could be a topic for future studies.

The authors mentioned that as ref#2 support the absence of spiral ganglion neuron expression, their CAP latency results cannot be due to a function in these cells. However, I could not find the supporting data in the reference. Imaging at high resolution (Fluorescence and EM) the hair bundles after DMSO and drugs would be important to ensure that there is no dramatic morphological defect induced to the organ of Corti and the hair bundle.

Khan et al (ref 2) used monoclonal antibodies to examine the expression of radixin in the organ of Corti. While they do not specifically comment on the neurons, they write that radixin expression was consistent with the one found by Kitajiri et al (2004), who observed intense radixin labeling only in the stereocilia. Had neuronal labeling been observed, we assume the authors of these previous papers would have said so (Kitajiri et al commented on other structures within the cochlea, such as the blood vessels of the stria). However, we agree that there is slight uncertainty about this important point, and we therefore examined our slides again, this time looking for expression in the nerve endings near inner hair cells. None was found.

We acknowledge that it is important to ensure that drugs or DMSO do not induce morphological defects in hair bundles. New high-resolution imaging data show no apparent morphological defects in stereocilia after drug application (the time span of these experiments is about 45 minutes which is the time window where the physiological data were acquired). High-resolution immunofluorescence images of the hair bundles were also acquired where the sensory epithelium was treated with drugs or DMSO via perfusion through the round window for 1 and 1/2 hr at room temperature. These new imaging data are included in the paper.

For the hair bundle motion, according to Fig 3C and F, the range of motion observed in control can vary a lot, and the authors had to normalize the values. What is not clear to me is how they calculated the Standard Deviation of these ratio, as it is not a straight forward thing (propagation of error) and it is crucial for the downstream statistical tests, especially because the differences are minimal.

We should have provided a better explanation of the procedures used when normalizing the data, and the methods section and figure legends were updated to provide such explanations.

In brief, each experiment starts by imaging bundle movements 3 - 5 times during a 20 – 30 minute time period. This is done to ensure we have a stable response before any drug is applied. Of course, no real experiment can produce exactly the same response on each trial. Since we normalize the data to the first measurement in each experiment, this means that there will be a standard deviation even for the normalized data.

For the electromobility measurements, the raw data presented (Fig 3 J) I can see only a minor difference upon DX-52-1 drug, and the “Before treatments” images looks very different from the

similar experiment with PAO (Fig 5H). Again, the importance of how the SD was determined is crucial for the statistical tests.

It is challenging to visualize the electromotility in a static image, because the maximum electromotile response that can be achieved is on the order of 200 nm. This is close to resolution limit of a confocal microscope, which means that two overlapping images can hardly be separated if the motion is at this magnitude. However, this problem does not apply to moving structures. This amplitude of motion is easily detected by optical flow algorithms, and it is easy to see if the images are assembled into a movie. Nevertheless, we think that the DX-52-1 images show a clear increase in electromotility. In the PAO experiment, the amplitude of the electromotility was smaller to begin with, and there was less of a change after PAO. Hence, the images look different.

For the FRAP data of Fig. 3, I don't understand how the half-recovery time can be compared between the conditions while the amount of bleached fluorescence vary so much. This could indicate that the recovering lipid pools could contribute differently between conditions. However, this seems appropriately controlled in the PAO experiment shown Fig. 5K. A good internal control could be to perform the experiment on the hair cell somatic membrane, that according to their model should not be affected.

Additional FRAP experiments were performed and are included in the supplementary data. Neither neurons or the cell bodies of the outer hair cells showed significant changes after DX-52-1.

REVIEWERS' COMMENTS:

Reviewer #1 (Remarks to the Author):

The authors have done an excellent job revising this paper and I am satisfied that all my concerns have been addressed. The manuscript text of the study is also significantly improved and the changes to the figures have made the results easier to understand.

I am happy that the PAO data remains in the main paper with the revisions to the text that have been done.

Reviewer #2 (Remarks to the Author):

The authors have submitted a revised manuscript that is much improved. I now find it to be suitable for publication.